# Building Extraction from Very-High-Resolution Remote Sensing Images Using Semi-Supervised Semantic Edge Detection

**Liegang Xia \*, Xiongbo Zhang, Junxia Zhang, Haiping Yang and Tingting Chen**

College of Computer Science and Technology, Zhejiang University of Technology, Hangzhou 310023, China; 2111812141@zjut.edu.cn (X.Z.); 2111912201@zjut.edu.cn (J.Z.); yanghp@zjut.edu.cn (H.Y.); 2111912209@zjut.edu.cn (T.C.)
* Correspondence: xialg@zjut.edu.cn; Tel.: +86-571-8529-0027

**Abstract:** The automated detection of buildings in remote sensing images enables understanding the distribution information of buildings, which is indispensable for many geographic and social applications, such as urban planning, change monitoring and population estimation. The performance of deep learning in images often depends on a large number of manually labeled samples, the production of which is time-consuming and expensive. Thus, this study focuses on reducing the number of labeled samples used and proposing a semi-supervised deep learning approach based on an edge detection network (SDLED), which is the first to introduce semi-supervised learning to the edge detection neural network for extracting building roof boundaries from high-resolution remote sensing images. This approach uses a small number of labeled samples and abundant unlabeled images for joint training. An expert-level semantic edge segmentation model is trained based on labeled samples, which guides unlabeled images to generate pseudo-labels automatically. The inaccurate label sets and manually labeled samples are used to update the semantic edge model together. Particularly, we modified the semantic segmentation network D-LinkNet to obtain high-quality pseudo-labels. Specifically, the main network architecture of D-LinkNet is retained while the multi-scale fusion is added in its second half to improve its performance on edge detection. The SDLED was tested on high-spatial-resolution remote sensing images taken from Google Earth. Results show that the SDLED performs better than the fully supervised method. Moreover, when the trained models were used to predict buildings in the neighboring counties, our approach was superior to the supervised way, with line IoU improvement of at least 6.47% and F1 score improvement of at least 7.49%.

**Keywords:** semi-supervised; semantic edge detection; building extraction; deep learning; very-high-resolution image



## 1. Introduction

In recent years, machine learning and image processing techniques have been using remote sensing images to mine abundant information for urban planning, change detection, disaster assessment and other fields. Taking earthquake monitoring as an example, soft computing techniques complete seismic vulnerability assessments of existing buildings, which mitigate postquake effects [1]. Meanwhile, many approaches have been developed to complete the task of rapidly assessing damage [2–4], which plays an important role in rescue and recovery missions. Pixel-based change detection is also widely used to accurately analyze the changes of destroyed houses [5]. These methods relying on computer technology are effective, which can extract building damage information in a very short time and greatly reduce labor costs. With the rapid development of remote sensing technology and the increasingly large observation system, remote sensing can provide detailed data for building monitoring [6]. Very-high-resolution (VHR) images have abundant spectral information, spatial characteristics and geometric characteristics [7]. However, artificial

processing is time-consuming and expensive and generates discrepancies in statements. How to separate buildings from complex backgrounds automatically and efficiently from VHR images has always been a hot topic in the remote sensing field [8].

For decades, scholars have been trying to automatically extract buildings from high-resolution remote sensing images. To better adapt to building extraction, researchers have made some targeted improvements to traditional algorithms [6,9–14]. However, these algorithms all have limitations at different levels, and the extracted building boundaries always have problems with incomplete boundaries and low extraction accuracy [15], and it is difficult to get good performance in further analysis of urban spatial planning. As deep learning based on convolutional neural networks (CNNs) achieves great effects in computer vision [16–18], an increasing number of scholars have tried to apply deep learning to extract buildings from VHR images. A faster edge region convolutional neural networks (FER-CNN) algorithm was proposed to achieve high accuracy building detection and classification, demonstrating great resistance to shadows [7]. Lu et al. [19] used the most peripheral constraint conversion algorithm to generate a superior dataset and trained the dataset with a richer convolutional features (RCF) network. The prediction building edges became accurate and complete after refining according to a geometric morphological analysis of the topographic surface. These CNNs' dependent approaches show superior performance in processing complex VHR remote sensing images. However, training a well-performing deep learning model with a fully supervised method requires large-scale manually annotated datasets. Differences between datasets of different resolutions and different regions cannot be avoided [20], and the model's generalization ability towards different datasets is not strong enough, showing instead a strong sample dependence.

The absence and difficulty of obtaining manual annotation datasets with the boundaries of buildings require more efforts to identify and draw [20] them than for roads, water bodies and woodlands, which drives scholars to explore ways to reduce the need for annotation datasets. The semi-supervised classification method can properly handle training sets with large amounts of unlabeled data and small amounts of labeled data [21]. Through an iterative process, the self-labeling method obtains a large predictive dataset and accepts the correctness, thus expanding the labeled dataset [22]. Significant progress has been made in many improved methods, such as Co-Forest [23], self-training nearest neighbor rule using cut edges (SNNRCE) [24] and weighted ternary decision structure (WTDS) [25].

However, those self-labeling methods are mostly based on handcrafted features, which may lead to misclassification. The learning model will quickly deviate from the actual model in the process of autonomous learning [26]. Combining CNNs and semi-supervised learning is a effective way to slow down the deviation rate. A novel dual-strategy sample selection cotraining algorithm provided the complementary cues of the spectral and spatial features for the residual networks (ResNets) to learn the classification information from unlabeled images [21], achieving great improvement on the Indian Pines dataset. Wu et al. [27] used deep convolutional recurrent neural networks (CRNNs) to label abundant unlabeled images and obtain high-quality classification pseudo-labels with a nonparametric Bayesian clustering algorithm. At present, most research is focused on simple datasets or classification tasks, and the superiority of semi-supervised methods has not been reflected in complex scenarios.

Therefore, we designed a semi-supervised deep learning method based on the semantic edges of buildings' roofs, aiming to train a pre-model by using a small number of labeled building samples and then self-label abundant unlabeled samples to obtain pseudo-labels [28]. We accepted the accuracy of these selected labels and formed an extended training dataset together with the manually labeled samples. Based on the pre-model, the extended dataset is used for iterative training to obtain the final fine-tuned model. Finally, through testing on the original dataset and other regional datasets, the superiority of the proposed method over the fully supervised method of the same scale is verified, and shows the stronger generalizability in other regions.

The main contributions of this paper are as follows: 1. Semi-supervised learning is introduced into the extraction of building roof edges in high-resolution remote sensing images, which verifies that the semi-supervised method has a better extraction effect than the fully supervised method under a similar sample amount. 2. By testing in the neighboring areas of the initial dataset, it is verified that this method has stronger generalization ability in other regions, which provides an idea for generalization ability based on semantic edges. We provide our codes publicly via the GitHub platform and provide our dataset publicly via the baiduyun platform to facilitate the reproduction of our results (see link in Data Availability Statement).

The rest of this article is organized as follows: The second section reviews the previous work related to semi-supervised learning and the deep neural networks for building edge extraction. The third section introduces in detail the semi-supervised learning framework to extract building edges using pseudo-labels and the neural network designed to complete edge detection. The fourth section introduces the dataset and experimental results and discusses the experimental results. The fifth section summarizes the thesis and provides some suggestions for future work.

## 2. Related Work

### 2.1. Building Extraction from VHR Images

There have been various traditional algorithms and theories of building detection, and their extraction effects have many problems in practical applications [19]. The good performance of CNNs in object classification, object extraction, semantic segmentation and edge detection has attracted an increasing number of researchers to try building extraction methods based on deep learning. Ji et al. [29] developed a scale robust CNN structure to extract buildings from high-resolution aerial and satellite images. Two dilated convolutions are used on the first two lowest-scale layers for enlarging the sight-of-view and integrating the semantic information of large buildings, and a multi-scale aggregation strategy is applied for improving segmentation accuracy. Hu et al. [15] took every building as an independent individual recognition using mask scoring R-CNN and added a mask intersection-over-union (IoU) head to score the mask, achieving better building extraction effects than mask R-CNN. Lu et al. [19] used RCF to obtain the edge strength map of a building and combined the geomorphological concept to optimize the edge of the building. Reda et al. [7] proposed the FER-CNN algorithm to improve the accuracy of building detection and classification and used the Ramer-Douglas-Peucker (RDP) algorithm to correct the shape of the detected buildings. The purpose of segmentation is not only satisfied with pixel-level assistance but also lies in object identification with accurate boundaries [14].

### 2.2. Deep Learning-Based Edge Detection

Edge extraction is one of the core challenges in computer vision. With the continued success of edge detection algorithms based on CNNs, their performance has far exceeded that of traditional computer vision methods [30]. The holistically nested edge detection (HED) network automatically learns rich hierarchical feature types under the guidance of deep supervision, which effectively solves the blurred edge problem of natural images [16]. Considering that HED adopts CNN features from only the last layer of each convolutional stage, rendering it unable to make full use of the rich feature hierarchy of CNNs, the RCF network uses a fully convolutional network (FCN) to effectively combine features from all the convolutional layers and make full use of the multi-scale and multi-level information of the target [31]. Dynamic feature fusion (DFF) recognizes the gain of multi-scale feature fusion on semantic edge detection and adaptively assigns appropriate fusion weights to different images and positions through the weight learner, thus more accurate and clearer generating edge prediction results than the fixed-weight fusion method [32]. By introducing a bidirectional cascade structure, the BDCN provides layered edge supervision of each layer on its specific scale and uses a scale enhancement module (SEM) to generate multi-

scale features, enriching the multi-scale representation of shallow network learning and improving the detection effect of target edges at different scales [33]. However, traditional deep learning methods often adopted a large number of manually labeled samples to ensure the good performance of deep learning, which is time-consuming and expensive. It is importance to reduce the dependence of networks on labeled samples.

### 2.3. Deep Learning-Based Semi-Supervised Method

The contradiction between the time-consuming and laborious labeling work and the sample demand of deep learning promotes the continuous development of semi-supervised learning. Fang et al. [21] proposed a semi-supervised deep learning framework based on residual networks (ResNets). The framework used a dual-strategy sample selection cotraining algorithm that fully utilizes complementary cues of spectral and spatial characteristics in hyperspectral image (HSI) classification, which successfully guides ResNets to learn from unlabeled data. Wu et al. [27] treated each hyperspectral pixel as a spectral sequence, used deep convolutional recursive neural network (CRNN) to classify HSIs and proposed a nonparametric Bayesian clustering algorithm—the constrained Dirichlet process mixture model (C-DPMM)—for semi-supervised clustering to produce high-quality pseudo-labels. Li et al. [34] proposed a self-cyclic uncertainty pseudo-label, which is generated by cyclically optimizing encoders in an FCN with self-monitoring subtasks, can be regarded as an approximation of the uncertainty estimation obtained by combining multiple models and significantly reduces the inference time. Han et al. [26] proposed a semi-supervised production framework, combined with deep learning features, self-labeling techniques and discriminative evaluation methods, to learn useful information from unmarked samples, significantly reduced the inference sets and completed the tasks of scene classification and set annotation. The classification task can make use of the information of VHR remote sensing images to some extent, but it cannot clearly indicate the position of buildings in the image, and it is completely insufficient in building extraction.

### 3. Methodology

In this section, we will describe the proposed approach for semantic edge extraction of buildings based on semi-supervised learning. We assume that limited labeled dataset L and a large-scale unlabeled high-resolution remote sensing dataset U are given. The designed framework is shown in Figure 1, in which the edge extraction algorithm based on deep learning is introduced to automatically mark unlabeled data, and the semantic segmentation network D-LinkNet is modified to accomplish the task of edge extraction. Now, we will introduce the two important components of this approach.

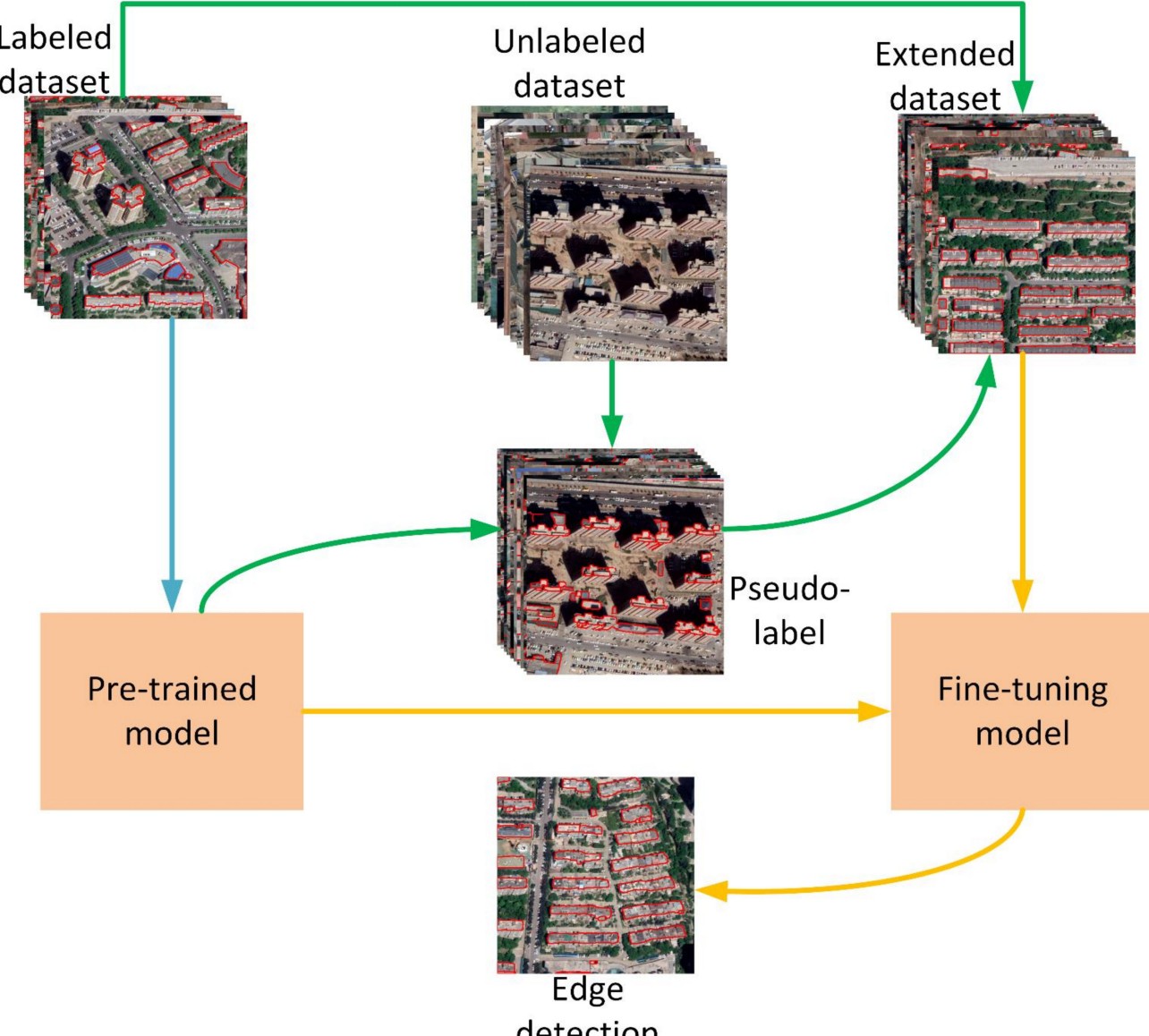

**Figure 1.** Edge-detection-based semi-supervised framework. A labeled dataset is used to train a pre-trained model, which predicts an unlabeled dataset to get pseudo-labels. The labeled dataset and pseudo-labels are combined to form an extended dataset, which has abundant labeled images. A fine-tuning model is trained with the extended dataset based on pre-trained model.

### 3.1. Network Architectures

Additional information to locate building locations and infer spatial locations can be achieved through semantic segmentation or edge detection. Semantic segmentation often ignores the interaction between object bodies and object edges, and its performance on boundary details is not satisfactory. Edge detection pays more attention to edge problems and involves lower-level semantic information. When making pseudo-labels of buildings with edge detection, the predicted results are more consistent with real building boundaries. Therefore, we modified D-LinkNet to adapt edge detection for extracting the boundaries of buildings in images.

D-LinkNet is a semantic segmentation neural network that adopts an encoder-decoder structure, dilated convolution and a pre-trained encoder [35]. The dilated convolution with skip connections in the center part is a useful part that adjusts the receptive fields of feature points without decreasing the resolution of the feature maps. The encoder part uses

a residual block with shortcut connections to make the network easier to optimize, which solves the degradation problem of deep neural networks.

As shown in Figure 2, our network is divided into two parts, A and B. Part A keeps the encoder part and center part of D-LinkNet. At each stage of part A, we use transposed convolutional layers to perform upsampling, which restores the resolution of the feature map to 448 × 448. Then, the network calculates the loss in each stage for more accurate edge location information. The deep convolutional layer has richer semantic information, while the shallow convolutional layer has richer edge details. To better realize semantic edge detection, the network adopts multi-scale fusion to identify the target building and locate the edge information more accurately.

Our loss function based on [16] can be formulated as

$$L_{side}^{m} = -\beta \sum logPr(y_j = 1) - (1 - \beta) \sum logPr(y_j = 0) \tag{1}$$

where $\beta = |Y_-|/|Y|$ and $1 - \beta = |Y_+|/|Y|$. $|Y|$ denotes the edge map. $|Y_-|$ and $|Y_+|$ denote the positive sample set and the negative sample set, respectively. $Pr(y_j = 1)$ is the standard sigmoid function, which computes the activation value at pixel y.

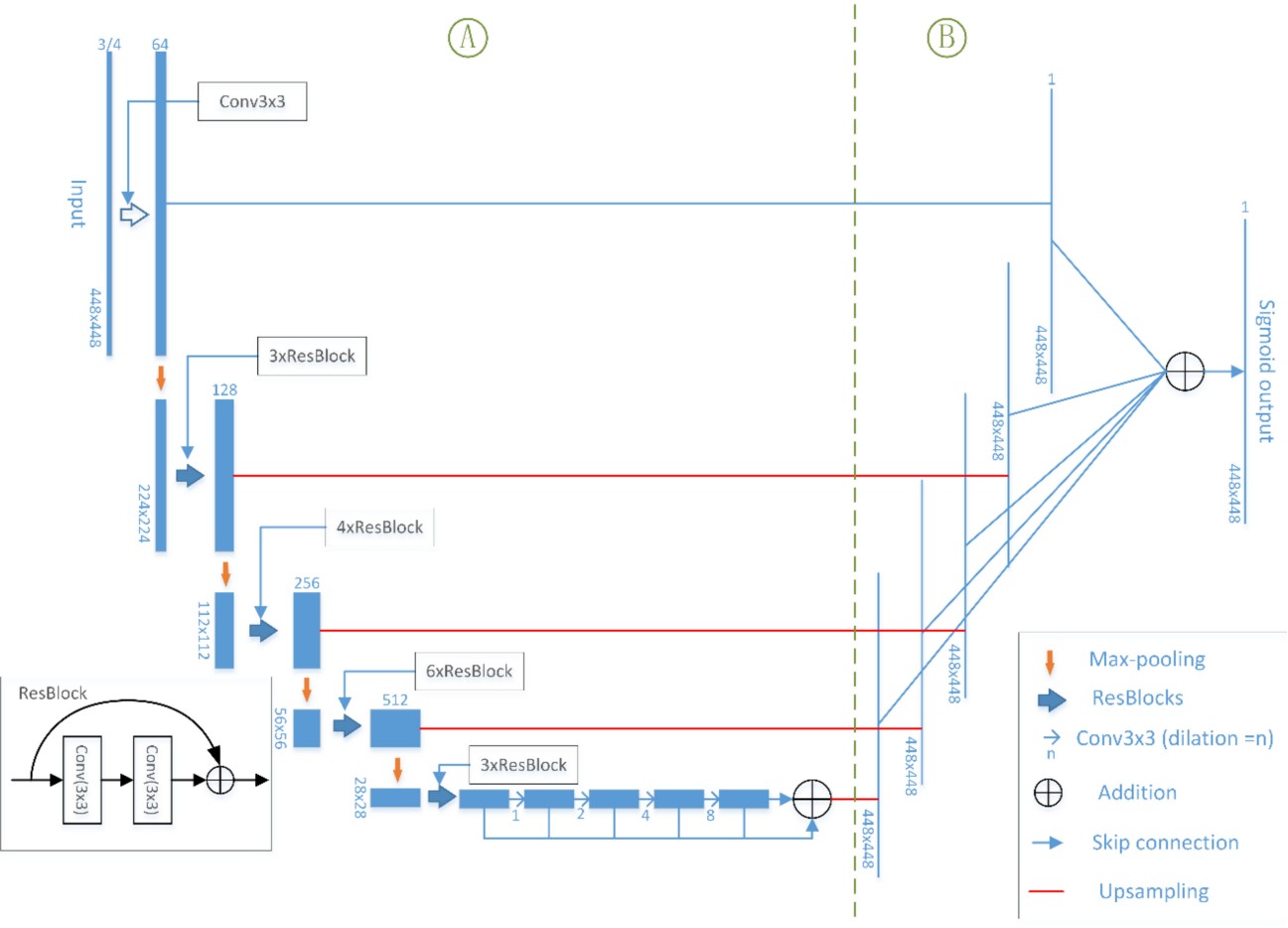

**Figure 2.** Our network for detecting the building edge, which is modified from D-LinkNet. D-LinkNet is composed of an encoder, center part and decoder. Part A in our network retains the original encoder and center part from D-LinkNet for effectively extracting the multi-scale features in the image. Part B uses multi-scale fusion to replace the decoder, has a wide receptive field and keeps the ability of spatial detail representation. For each stage of the encoder, we use the transposed convolution layer for upsampling and restoring the resolution of the feature image to 448 * 448. Then, the network calculates the loss function of each stage to obtain more accurate edge location information.

*3.2. Training Process*

The training process of this paper mainly includes three aspects: training the pre-model, updating the sample pool and fine-tuning the model. We use the small-scale marked sample set L to train the pre-model and carry out automarking on the large-scale unlabeled high-resolution remote sensing dataset U. Since L and U are images taken from the same period and same region, we believe that the correlation of architectural semantic information between the two datasets is sufficient for the pre-model to generate acceptable pseudo-label information. To select labels with complete boundaries and high confidence, a selection method based on building edge integrity was used and will be introduced in the next section. The selected pseudo-label information and its corresponding image are added to L to form a new extended label sample set L', which is used for iterative training of the pre-model to obtain the final fine-tuned model.

## 4. Experiment and Results

*4.1. Dataset*

In this paper, we used Google Earth images of level 18 with a spatial resolution of 0.522 m, taken in Beijing, the capital of China. It is a typical representative of urban architecture, including a variety of typical buildings that present different styles in VHR remote sensing images. It is highly representative of the problem of building boundary extraction involved in this study. The remote sensing images in this dataset covered nine districts, which were divided into three parts. The first part selects 344 images for training and testing from Chaoyang, Haidian, Dongcheng and Xicheng and is divided into a training set (80%) and a test set (20%). The second part contains 1564 unlabeled images from the same area as the first part. In the third part, 288 images were selected from five districts around the first part for the generalization ability test. In addition, to verify the role of the number of samples in our experiment, we train our models based on different scales (25%, 50%, 75%, 100%) of the training set in the first part. The location relationship is shown in Figure 3. All parts of the data are 512 * 512 pixels in size. As shown in Figure 4, we use ArcGIS 10.2 to draw labels for the first and third parts according to the roof of the building. The dataset covers buildings of different heights, sizes, types and densities, such as tall commercial buildings, low and dense rural houses, neatly planned modern communities and sporadic single-family buildings. The first part of the samples is used in training the pre-model. The second part of the samples is added into the first part after prediction by the pre-model. The two portions of samples form an extended dataset for training the fine-tuning model. The third part of the samples is only used for the generalization ability test.

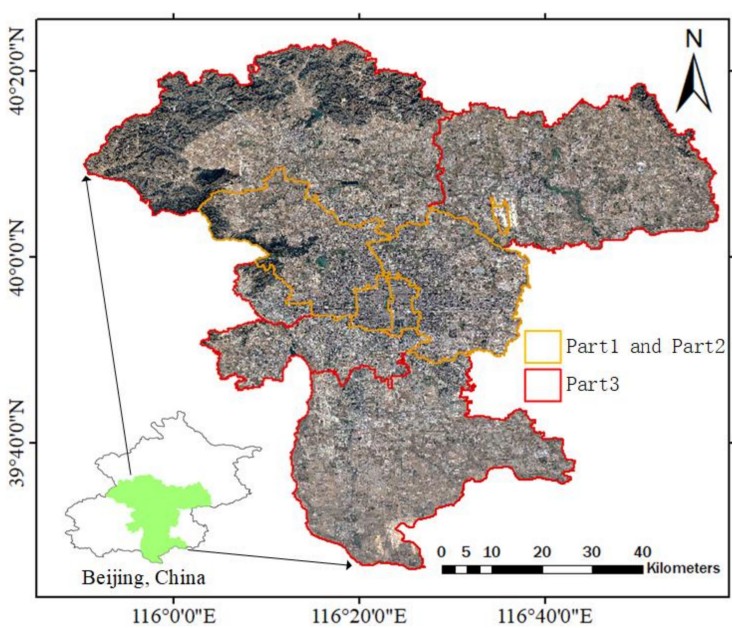

**Figure 3.** The first and second parts of the sample are both taken from the yellow area. The first part is the training and testing set. The second part is the unlabeled dataset. The third part is taken from the blue area for the generalization ability test.

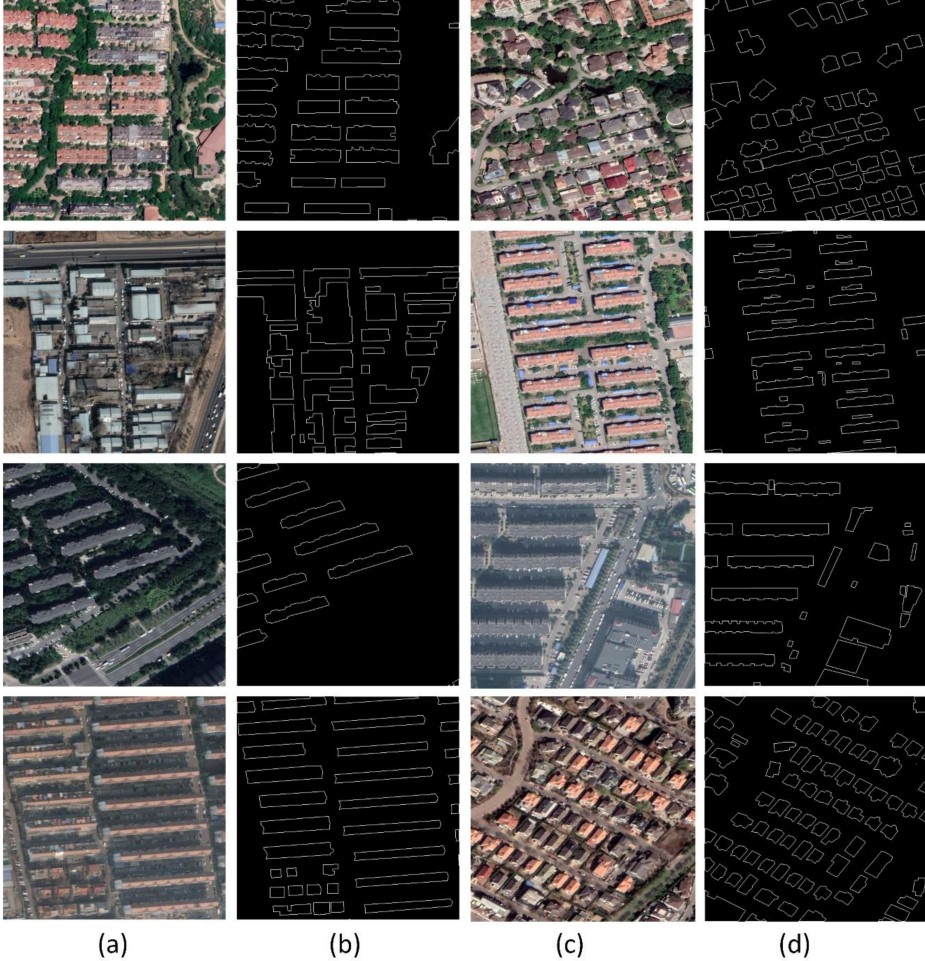

(a)    (b)    (c)    (d)

**Figure 4.** Example images from the building dataset taken in Beijing from Google Earth. (**a**) Original image for training and testing, (**b**) edge labels of (**a**), (**c**) original image for generalization ability testing and (**d**) edge labels of (**c**).

*4.2. Training Details*

In our experiments, PyTorch was applied as the deep learning framework, and all the models were trained with 2 NVIDIA GeForce 1080 Ti GPUs. All the networks were trained with the Adam optimizer. The learning rate was originally set to $1 \times 10^{-3}$ and reduced by five at three times because the training loss decreased slowly. The batch size during the training phase was fixed at four. For fairness, it took 200 epochs to train the pre-model and fine-tuned model and took 400 epochs for the fully supervised model. To avoid overfitting, we performed data augmentation, including random cropping, horizontal flipping, vertical flipping, diagonal flipping and rotation, to enlarge our dataset. The cropped image size was 448 * 448, while the original size was 512 * 512.

*4.3. Evaluation*

We use the IoU and F1 score [36] to evaluate the performance of the proposed method in this paper. The IoU is used to measure the degree of correlation between the real background and the predicted results; the higher the value of the IoU is, the stronger the correlation is. The calculation formula is as follows,

$$IoU = \frac{A \cap B}{A \cup B} \tag{2}$$

where A represents the predicted area, and B represents the real label area.

There are three different ways to calculate the IoU in this paper. 1. The whole IoU calculates the overlap between the predicted value and the real label on the whole image. 2. The single IoU is the average IoU of each building and focuses on the number of buildings. 3. We adopt the line IoU to describe the correlation between the real and predicted building boundaries. The real and predicted building boundaries are expanded with the same expansion kernel, and then the line IoU of the building boundary is calculated.

The F1 score is an index used to measure the accuracy of the binary classification model. To calculate the F1 score, it is necessary to calculate the precision and recall of each prediction. In the following formula, TP (true positives) is the number of positive pixels that are correctly recognized as positive pixels. TN (true negatives) is the number of negative pixels that are correctly recognized as negative pixels. FP (false positives) is the number of negative pixels that are wrongly recognized as positive pixels. FN (false negatives) is the number of positive pixels that are incorrectly recognized as negative pixels. The precision is the percentage of true positives out of the positive pixels identified. The recall is the proportion of all positive pixels in the test set that are correctly identified as positive pixels. The F1 score is the harmonic mean of the precision and recall, relying on the idea that the accuracy and recall are equally important.

$$Precision = \frac{TP}{TP + FP} \tag{3}$$

$$Recall = \frac{TP}{TP + FN} \tag{4}$$

$$F1 = 2 * \frac{Precision * Recall}{Precision + Recall} \tag{5}$$

*4.4. Results*

4.4.1. Effect of the Sample Scale in the Supervised Method

For verifying the influence of the sample scales on our proposed network, we trained the network with the fully supervised method on an increased number of training images, respectively, in the proportions of 25%, 50%, 75% and 100%. The results are shown in Table 1 and Figure 5; as the high-quality training samples increase, CNNs learn more types of buildings and more comprehensive image features, and all the measures of precision estimation are in an upward trend until it becomes stable. The model trained by 100%

training samples shows competitive performance with line IoU (kernel = 3) improvement by 10.51% and F1 score improvement by 12.68% with the results from the model trained by 25% samples. Under this premise, we will explore the performance of increasing the number of unlabeled samples in the next section.

**Table 1.** Precision evaluation results for the effect of the sample scale in the supervised method.

| Sample Scale | F1 Score | Single IoU | Whole IoU | Line IoU (Kernel = 3) | Line IoU (Kernel = 5) |
|---|---|---|---|---|---|
| 25% | 0.7151 | 0.5723 | 0.5556 | 0.4038 | 0.4891 |
| 50% | 0.7828 | 0.6380 | 0.6311 | 0.4553 | 0.5515 |
| 75% | 0.8388 | 0.6842 | 0.7046 | 0.4825 | 0.5871 |
| 100% | 0.8419 | 0.7065 | 0.7111 | 0.5089 | 0.6152 |

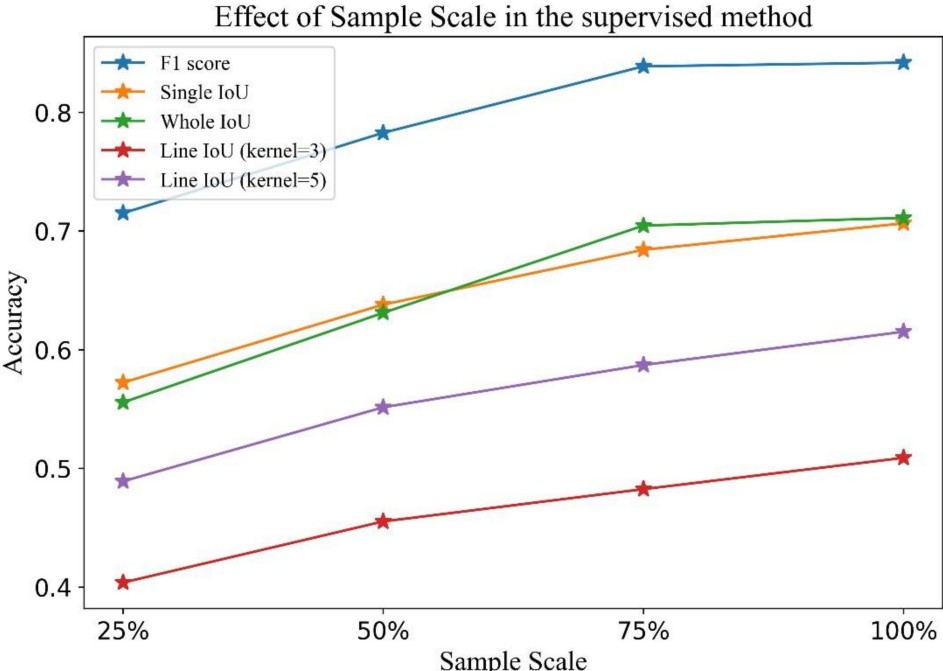

**Figure 5.** Precision evaluation results for the effect of the sample scale in the supervised method. This figure contains all the results obtained by our proposed network with the fully supervised method.

### 4.4.2. Effect of Edges in SDLED Prediction

In an attempt to explore the significance of edge pseudo-labels, we train our network with different scales (50%, 75%, 100%) of datasets for generating pseudo-labels of different quality, which form extended datasets together with the manually labeled samples for training the final fine-tuned model. The results of the semi-supervised method are compared with the fully supervised method that uses the 100% manually labeled samples, as shown in Figure 6. When 50% of the labeled samples are used, the semi-supervised method is better than the fully supervised one in F1 score and polygon IoU. To further verify the effectiveness of the edge detection network in the semi-supervised method, additional edge detection networks' BDCNs are selected as comparative experiments, and the results from our network are recorded in Table 2. In addition, to facilitate the visualization and interpretation of the results of BDCN and our network, we show some predicted images by networks and the final refined results of the semi-supervised and fully supervised methods in Figure 7.

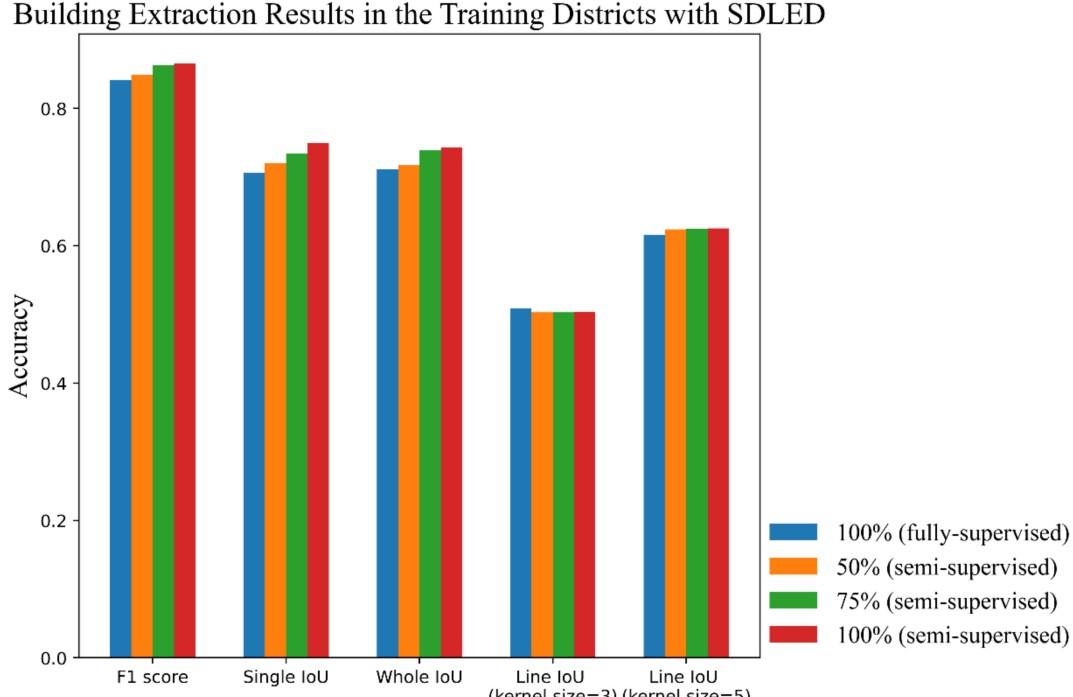

**Figure 6.** Precision evaluation results for the effect of edges in SDLED.

**Table 2.** Precision evaluation results for the effect of edges in SDLED.

| Network | | Sample Scale | | | |
|---|---|---|---|---|---|
| | | 100% (Fully) | 50% (Semi) | 75% (Semi) | 100% (Semi) |
| BDCN | F1 score | 0.5362 | 0.6267 | 0.6613 | 0.7014 |
| | Single IoU | 0.4043 | 0.4950 | 0.5254 | 0.5509 |
| | Whole IoU | 0.3941 | 0.4866 | 0.5281 | 0.5724 |
| | Line IoU (kernel = 3) | 0.3233 | 0.3337 | 0.3503 | 0.3676 |
| | Line IoU (kernel = 5) | 0.3849 | 0.4281 | 0.4487 | 0.4706 |
| Ours | F1 score | 0.8419 | 0.8490 | 0.8632 | 0.8650 |
| | Single IoU | 0.7065 | 0.7204 | 0.7346 | 0.7495 |
| | Whole IoU | 0.7111 | 0.7175 | 0.7391 | 0.7436 |
| | Line IoU (kernel = 3) | 0.5089 | 0.5030 | 0.5033 | 0.5037 |
| | Line IoU (kernel = 5) | 0.6152 | 0.6240 | 0.6243 | 0.6255 |

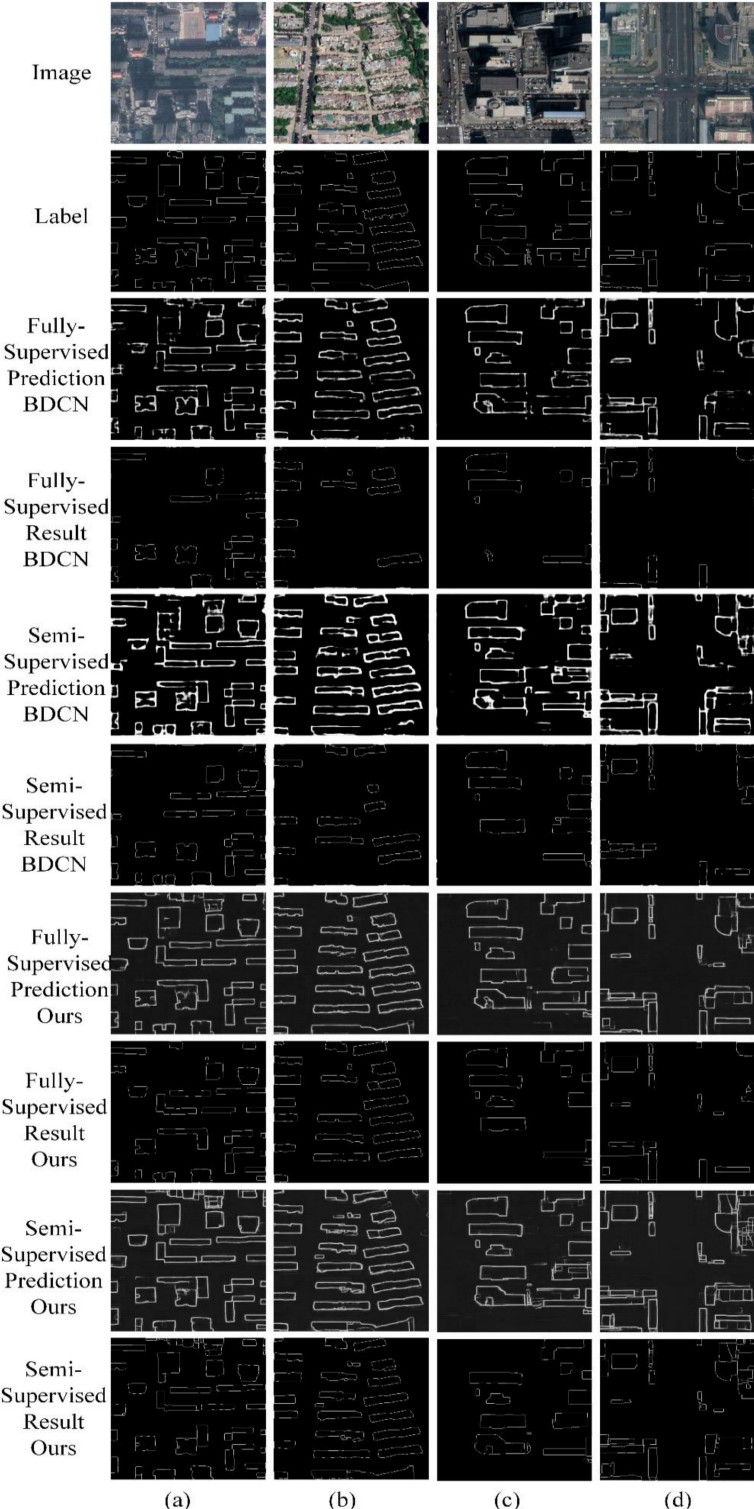

**Figure 7.** (**a**–**d**) are the results for fully supervised and semi-supervised building extraction in training distinctions. There are two edge detection models, BDCN and our model. The semi-supervised method uses the 50% labeled dataset and 100% unlabeled dataset, while the fully supervised method uses the 100% labeled dataset and 0% unlabeled dataset. The image is the original image taken in Beijing from Google Earth. The label is the ground truth of the image. Prediction is the edge strength map, which is the result of our network. The fully supervised and semi-supervised results are the complete edges, which, after post-processing, include binarization and thin, incomplete boundary elimination.

### 4.4.3. Generalization Ability Analysis in SDLED Prediction

Around the four training districts selected in Beijing, we selected five districts to analyze the generalization ability of the semi-supervised method. We use the model in the training districts to predict the samples in the neighboring districts, and the results are shown in Table 3 and Figure 8. Compared with the fully-supervised method, our approach achieves competitive performance with line IoU improvement of at least 6.47% and F1 score improvement of at least 7.49% for edge detection, while using half of the labeled samples. Compared with the training area, the values of all the evaluating indicators decreased obviously, but the improvements in the semi-supervised method more than the fully supervised method are magnified. The images of prediction and results are shown in Figure 9.

**Table 3.** Precision evaluation results for generalization ability analysis in SDLED.

| Sample Scale | F1 Score | Single IoU | Whole IoU | Line IoU (Kernel = 3) | Line IoU (Kernel = 5) |
|---|---|---|---|---|---|
| 100% (fully supervised) | 0.7013 | 0.5867 | 0.5573 | 0.3732 | 0.4787 |
| 50% (semi-supervised) | 0.7762 | 0.6969 | 0.6535 | 0.4379 | 0.5648 |
| 75% (semi-supervised) | 0.7867 | 0.7021 | 0.6634 | 0.4357 | 0.5667 |
| 100% (semi-supervised) | 0.7901 | 0.7216 | 0.6710 | 0.4452 | 0.5801 |

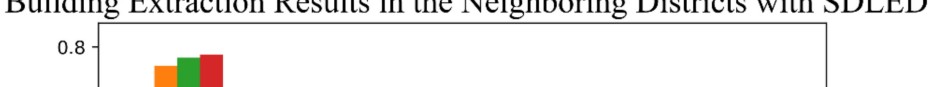

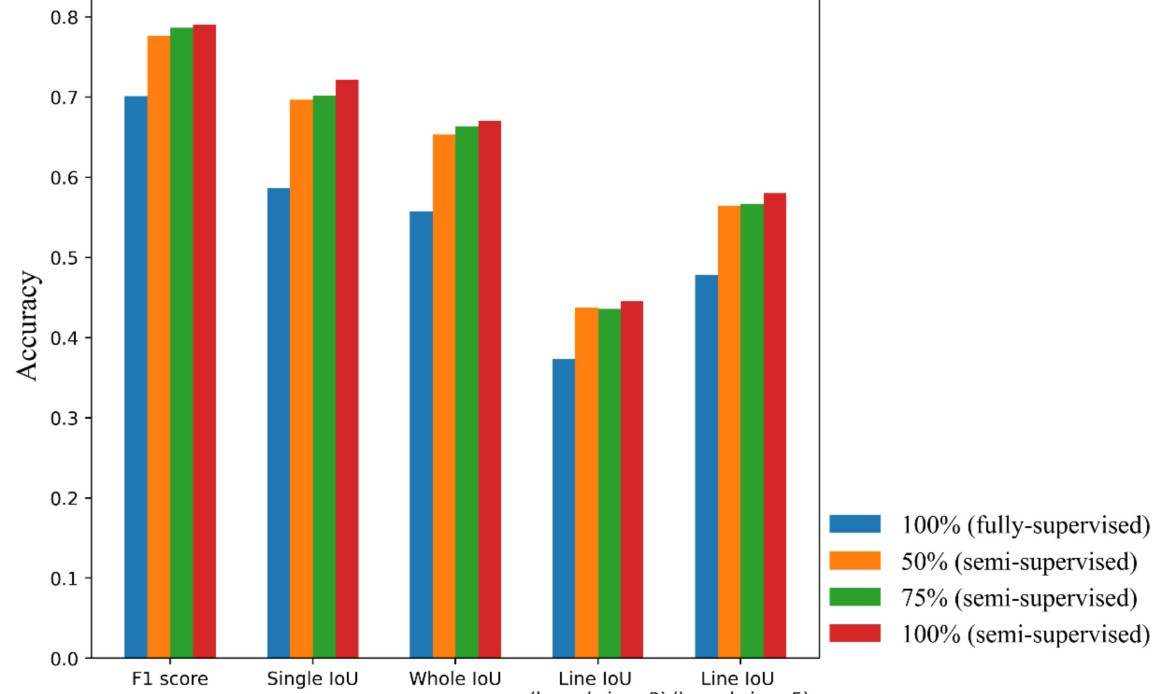

**Figure 8.** Precision evaluation results for generalization ability analysis in SDLED.

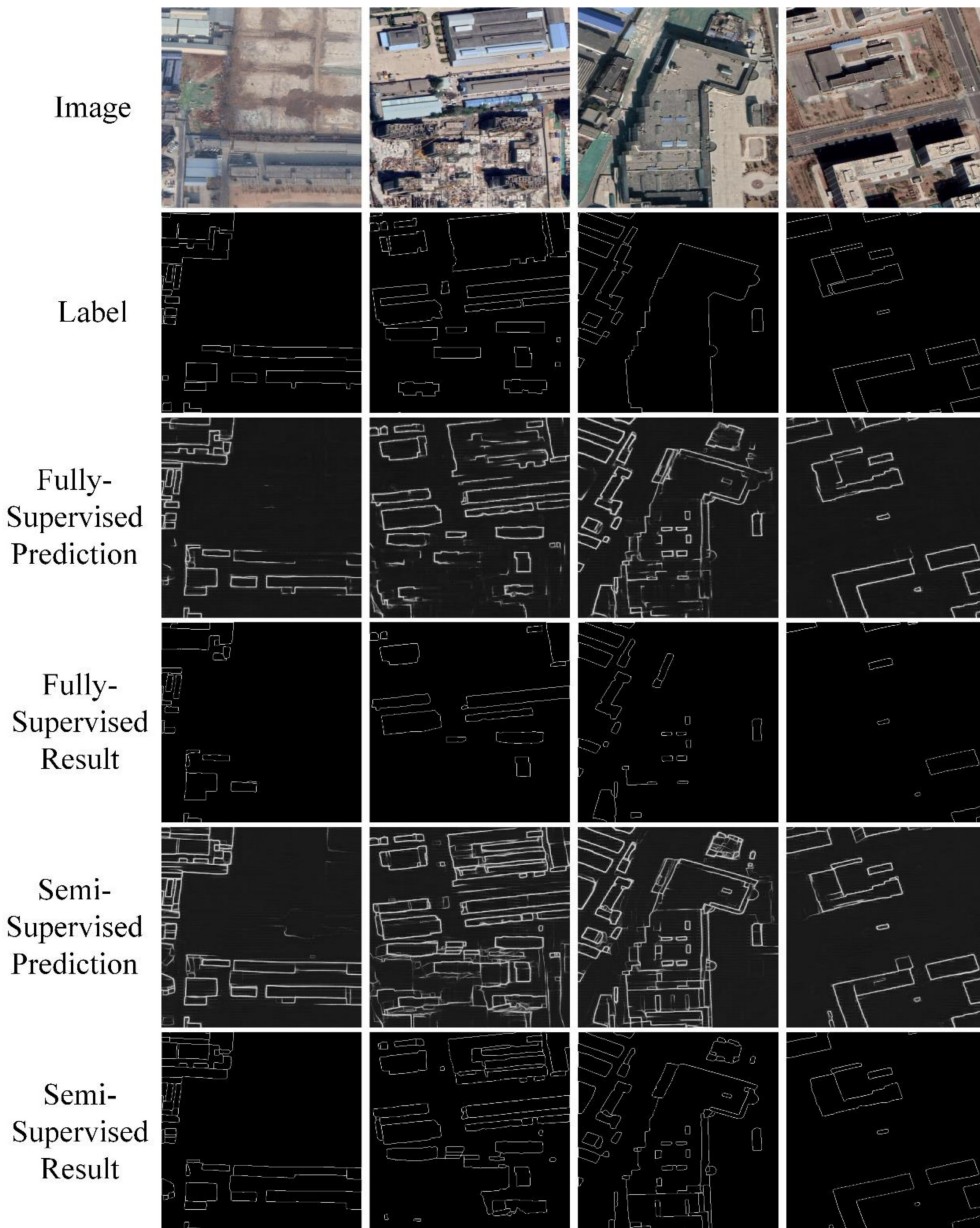

**Figure 9.** The results of building extraction for fully supervised and semi-supervised methods in neighboring distinctions. The semi-supervised method uses the 50% labeled dataset and the 100% unlabeled dataset, while the fully supervised method uses the 100% labeled dataset and the 0% unlabeled dataset. The image is the original image taken in Beijing from Google Earth. The label is the ground truth of the image. Prediction is the edge strength map, which is the result of our network. The fully supervised and semi-supervised results are the complete edges, which, after post-processing, include binarization and thin, incomplete boundary elimination.

### 4.4.4. Sample Selection Method Analysis in Semi-Supervised Prediction

In the pseudo-labels with different qualities, we found quite a few unacceptable prediction results. Therefore, we tried to screen high-quality labels by the degree of boundary integrity and test them according to an integrity threshold value of 0.7 and the top 80% of the pseudo-labels in positive order, which are compared with the nonselected semi-supervised method, as shown in Table 4 and Figure 10. However, there was no significant improvement in the accuracy between the two selection methods. By comparing the rejected pseudo-labels and training samples, we find that the building types not included in the manually labeled samples are those with poor edge prediction (Figure 11). The pseudo-label selection method regardless of building types is not suitable for this experiment.

**Table 4.** Precision evaluation results for sample selection mechanism analysis in the semi-supervised method. 50%, 75% and 100% are the scale of manually labeled samples. The sample selection method calculates the ratio of the complete boundary in prediction results to the real boundary to select higher-quality pseudo-labels. Unselected means all the pseudo-labels participate in training without selection. Threshold 0.7 means the method sets a threshold of 0.7 to remove incomplete samples. Top 80% means the top 80% of pseudo-labels with the highest ratio will be used for training.

| Sample Selection Mechanism | | Sample Scale | | |
|---|---|---|---|---|
| | | **50%** | **75%** | **100%** |
| Unselected | F1 score | **0.849** | **0.8632** | 0.865 |
| | Single IoU | 0.7204 | 0.7346 | 0.7495 |
| | Whole IoU | 0.7175 | **0.7391** | 0.7436 |
| | Line IoU (kernel = 3) | **0.503** | **0.5033** | 0.5037 |
| | Line IoU (kernel = 5) | **0.624** | **0.6243** | 0.6255 |
| Threshold 0.7 | F1 score | 0.8128 | 0.8618 | 0.8628 |
| | Single IoU | 0.7195 | 0.7361 | **0.7563** |
| | Whole IoU | 0.7087 | 0.7375 | 0.7398 |
| | Line IoU (kernel = 3) | 0.486 | 0.4917 | **0.5114** |
| | Line IoU (kernel = 5) | 0.6004 | 0.6129 | **0.6369** |
| Top 80% | F1 score | 0.8349 | 0.8515 | **0.8746** |
| | Single IoU | **0.7331** | **0.7416** | 0.7511 |
| | Whole IoU | **0.7176** | 0.722 | **0.7561** |
| | Line IoU (kernel = 3) | 0.495 | 0.4883 | 0.5088 |
| | Line IoU (kernel = 5) | 0.6129 | 0.613 | 0.6317 |

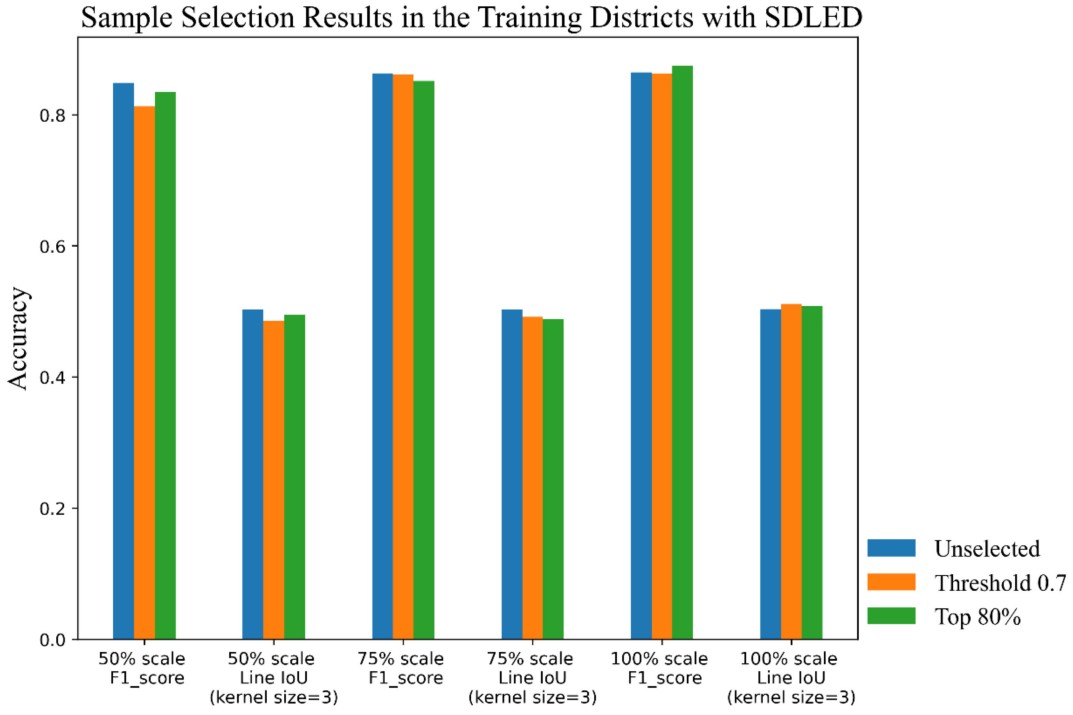

**Figure 10.** Precision evaluation results for sample selection mechanism analysis in SDLED. 50%, 75% and 100% are the scale of manually labeled samples. The sample selection method calculates the ratio of the complete boundary in prediction results to the real boundary to select higher-quality pseudo-labels. Unselected means all the pseudo-labels participate in training without selection. Threshold 0.7 means the method sets a threshold of 0.7 to remove incomplete samples. Top 80% means the top 80% pseudo-labels with the highest ratio will be used for training.

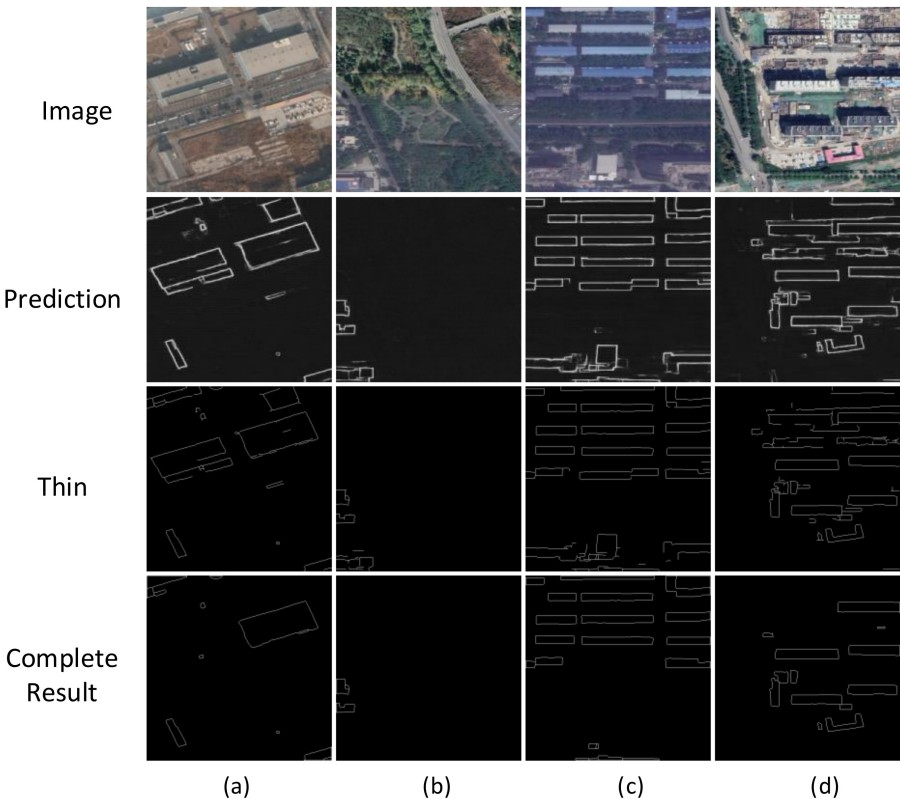

**Figure 11.** (**a**–**d**) are the results of building extraction for the fully supervised method with 50% samples in training distinctions. The image is the original image taken in Beijing from Google Earth. The prediction is the edge strength map, which is the result of the fully supervised method. The thin is the result of the prediction after binarization and thin. The complete result is the result of the thin after incomplete boundary elimination.

## 5. Discussion

This study aimed to research the applicability of the semi-supervised method in the edge detection network by comparing the performance of the semi-supervised and fully supervised methods on the Beijing dataset of building roofs. The results showed that increasing the number of samples could improve the effect of the edge detection network, whether the label is accurate enough or not. According to the results in Tables 2 and 3, with the information from the unlabeled samples, our semi-supervised method needed only half of the manually labeled samples to achieve better performance than the fully supervised method. The deep learning method based on neural networks has a serious dependence on the sample scale. Expanding the scale of datasets will enrich the feature information and context information of buildings. As shown in Figure 7, more complete buildings are obtained by our method than the fully supervised method as more context information is added. The quality of labeled samples will vary according to the level, status and understanding of different experts, which will provide inaccurate information to the neural network. However, the neural network can tolerate some error information in the acceptable range and still show a good detection effect. The inaccurate pseudo-labels generated by the semi-supervised method are also within the acceptable range of the network. Due to the expansion of the training images, which bring more abundant learning information, the ability of the model has been improved. From the perspective of line IoU (kernel = 3), which represents the fitting degree between the predicted edge and the real building boundary, with the addition of abundant pseudo-labels, the ratio of false information to correct information increased greatly, which led to the predicted edge deviating from the real boundary. Because the semi-supervised method of the extended dataset had more target information, more edges are found in the prediction results. After

the expansion of the 5-pixel range, the overlap with the real boundary increased, so the value of the line IoU (kernel = 5) was improved. By comprehensively evaluating a variety of indicators, we believe that inaccurate pseudo-labels in building edge detection are effective, which brings about a better detection effect. In addition, we compare our network with another edge detection network, BDCN. Although the performance of BDCN on boundary continuity is not satisfactory in our dataset, in comparing the semi-supervised method with the fully supervised method, the former showed the line IoU improvement of at least 1.04% and the F1 score improvement of 9.05%. The semi-supervised method is effective for the edge detection model to use in improving accuracy when only a small number of manually labeled samples are accessible.

To be of value in geographic and social applications, it is also important that the model performs well on a district that it has not been trained on. The results of the model acting on the neighboring untrained districts show that the semi-supervised method has stronger generalization ability than the fully supervised method. Different types of buildings emerge in the neighboring area, wherein the context information contained is not well grasped by the fully supervised method, so the performance of the prediction is unsatisfactory. However, with the addition of pseudo-labels and the expansion of the training set, the improvement in accuracy with the semi-supervised method is particularly obvious. Although there is a great deal of completely wrong edge information in the pseudo-labels that will affect the judgment of the model, the positive effect brought by the supplementary type and feature information is dominant, as shown in Figure 9; the result provides more accurate guidance for the identification of unknown targets.

In the study in Section 4.4.4, there was no significant improvement in the results by the pseudo-label selection method. This method only considers the integrity of the boundary and ignores the untrained building types, which is not comprehensive enough. Samples with unknown building features are more likely to have incomplete edge extraction and are easily excluded by the selection method. Moreover, removing the bad labels reduces the cognition of the model. Therefore, the method of optimizing the pseudo-label should be based on avoiding discarding as much sample information as possible. In the next step, we will consider using a postprocessing method to improve the quality of the prediction results to achieve the optimization.

## 6. Conclusions

To reduce the impact of training dataset shortages on deep learning training, in this paper, we propose a semantic edge detection approach for very-high-resolution remote sensing images based on semi-supervised learning. This is the first time that semi-supervised learning and edge detection neural networks have been combined for extracting building roof boundaries from VHR remote sensing images. This method uses a small number of labeled images to train the pre-model, which is used to guide the unlabeled datasets for generating pseudo-labels. The training dataset is expanded by manual labels and pseudo-labels. In particular, D-LinkNet was modified to improve the quality of the pseudo-labels. Results show that, with the same number of labeled samples, our method achieves significant improvement in the IoU and F1 score compared to the fully supervised method, and the precision on some indicators is close to or even better than that of the fully supervised method of several labeled samples. In addition, by testing in the surrounding areas, the improvement effect of our method is more obvious, which proves the generalization of our method and provides a new way of thinking for generalization learning. In future studies, we will further explore the impact of semi-supervised learning on generalization learning to improve the effect of labeled samples in the experimental area on the extraction of building edges in wider regions.

**Author Contributions:** L.X. and X.Z. designed and conducted the experiments and wrote the first draft. J.Z., H.Y. and T.C. guided this process and helped with the writing of the paper. All authors have read and agreed to the published version of the manuscript.

**Funding:** This work was supported in part by the National Key Research and Development Program of China under [Grant 2018YFB0505303], in part by the National Natural Science Foundation of China under [Grant 41701472] and Zhejiang Provincial Natural Science Foundation of China under Grant No. LQ19D010006.

**Institutional Review Board Statement:** Not applicable.

**Informed Consent Statement:** Not applicable.

**Data Availability Statement:** The paper provides the database used in the current study at baiduyun (Link: https://pan.baidu.com/s/1953urFhpLZAOEZbyniIAoQ, Extraction code: n7jf) until 2 June 2022 and the python code available online at GitHub (https://github.com/lovexixuegui/SDLED), accessed on 2 June 2021.

**Acknowledgments:** We would like to acknowledge the Imagesky in Suzhou for supporting the Google Earth data.

**Conflicts of Interest:** The authors declare no conflict of interest.

## Abbreviations

| | |
|---|---|
| BDCN | Bi-Directional Cascade Network |
| C-DPMM | Constrained Dirichlet process mixture model |
| CNN | Convolutional neural network |
| CRNN | Convolutional recursive neural network |
| DFF | Dynamic feature fusion |
| D-LinkNet | LinkNet with Pretrained Encoder and Dilated Convolution |
| FCN | Fully convolutional network |
| FER-CNN | Faster edge R-CNN |
| FN | False negatives |
| FP | False positives |
| GPU | Graphics processing unit |
| HED | The holistically nested edge detection network |
| HSI | Hyperspectral image |
| IoU | Intersection-over-Union |
| RCF | Richer convolutional features network |
| RDP | Ramer–Douglas–Peucker |
| ResNet | Residual network |
| SDLED | Semi-supervised deep learning approach based on the edge detection network |
| SEM | Scale enhancement module |
| SNNRCE | Self-training nearest neighbor rule using cut edges |
| TN | True negatives |
| TP | True positives |
| VHR | Very-High-Resolution |
| WTDS | Weighted ternary decision structure |

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
