# Peer review of "Building Extraction from Very-High-Resolution Remote Sensing Images Using Semi-Supervised Semantic Edge Detection"

_remotesensing, doi:10.3390/rs13112187_

Round 1
Reviewer 1 Report
The authors have addressed most of my comments, but please include the links to the code and dataset in your paper. If someone is interested in your work, it should be easy for them to find where the code and datasets are.
Author Response
Dear reviewer,
Thank you very much for your recognition of our paper and the valuable comments on the first submission concerning our manuscript entitled “Building Extraction from Very-High-Resolution Remote Sensing Images Using Semi-Supervised Semantic Edge Detection” (remotesensing-1076964).

Reviewer 2 Report
I have reviewed the revised and resubmitted manuscript having the title: Building Extraction from Very-High-Resolution Remote Sensing Images Using Semi-Supervised Semantic Edge Detection.
I have already reviewed the previous two version of the manuscript and have tracked the progress. For my side the last version had only that concern that the manuscript has still not yet provided enough extended discussion of the findings as a comparison of other published references.
I can now acknowledge that in it's recent version the authors have improved the manuscript and the updated Results and Discussion chapter suits more to the requirements of a leading scientific journal article.
From now I do not have more issues with the content of the manuscript.
Author Response

(The authors gave the same response as above.)

Reviewer 3 Report
Dear Authors
The manuscript entitled “Building Extraction from Very-High-Resolution Remote Sensing Images Using Semi-Supervised Semantic Edge Detection” investigated a semi-supervised deep learning approach based on the edge detection network (SDLED) performs better than other previous methods.
The submitted paper sounds to be interesting but requires further significant corrections and considerations to become an acceptable work on this journal.
- As you have many abbreviations, therefore please provide an abbreviation table that on MDPI format must be at the end of the paper, however you made it but is not complete.
- The abstract is too confusing and too short and does not present important points. It should be between 250 to 300 words and concisely mention the problems of previous works and novelties in this paper.
- Similarly, the introduction has very poor structure and lack of literature review. Usually in the chapter of introduction the background and needs of study of for example seismic risk assessment and mapping methods must be highlighted and prepare readers to go further. Then your second chapter should be literature review where you present an overview on the previous works and the main problem statements of work and how it can be improved or overcome on it. 4-Please provide more information about the selected location and data repository and how they have been collected.
- Please provide better quality/shape for figures (especially Fig 1 and 2) and provide a systematic format and shape and legend for them. They need more descriptions on the body of paper to explain it in detail not in the figure label.
- Results and discussion are not properly organized, and it must show the significant achievements of the proposed method and discuss each table and figure properly and in detail.
- It could be great if you do a comparison between your proposed method and some of the available or common other methods to show the efficiency of it.
- It would be helpful if you provide a general framework or flowchart that how others can implement or use your proposed method for their assessment purposes.
- In total, your conclusion is good, and you can discuss a bit again about the achievements and novelty of your proposed method.
- The paper needs a detailed proofreading and paraphrasing.
- Please make your tables and figures follow a same path and font size and colors and adjust them in a proper way, table 2 is totally disorganized.
- You did not highlight the problem statement, objectives and novelty of your proposed method; That is why increasing the background of literature review based on the recommended works can help in this manner.
- Please provide more information about the analysis rather than just F1 score. Show your confusion matrix and the parameters you can achieve from it.
- Overall, the main problem of your paper is the lack of literature review and you can present some new developed machine learning methods and image processing for damage detection, seismic vulnerability, and risk assessment of buildings to attract the attention of readers and show a wide view of your works. Below are some of the recent works, where I found them new and useful to add and make your paper much more interesting:
-ML-EHSAPP: a prototype for machine learning-based earthquake hazard safety assessment of structures by using a smartphone app
-Post-earthquake road damage assessment using region-based algorithms from high-resolution satellite images
-Earthquake Hazard Safety Assessment of Existing Buildings Using Optimized Multi-Layer Perceptron Neural Network
- Multi-Hazard and Spatial Transferability of a CNN for Automated Building Damage Assessment
- Automatic image-based event detection for large-N seismic arrays using a convolutional neural network
PPM-SSNet: An Pyramid Pooling Module-based Semi-Siamese Network for End-to-end Building Damage Assessment from Satellite Imagery -Seismic loss estimation software: A comprehensive review of risk assessment steps, software development and limitations
-Evaluation of Change Detection Techniques using Very High Resolution Optical Satellite Imagery
Author Response
Dear reviewer,
Thank you very much for your generous help with our paper and the reviewers’ valuable comments concerning our manuscript entitled “Building Extraction from Very-High-Resolution Remote Sensing Images Using Semi-Supervised Semantic Edge Detection” (remotesensing-1076964). We have carefully revised the paper according to these comments. The point-to-point response to the reviewers’ comments is listed below.
As the reply is a little long, you could read the word directly.
p.s. In the revised version, the red parts are the changes based on the reviewers’ comments. The line number mentioned in the following responses are all collected from our revised manuscript.
The manuscript entitled “Building Extraction from Very-High-Resolution Remote Sensing Images Using Semi-Supervised Semantic Edge Detection” investigated a semi-supervised deep learning approach based on the edge detection network (SDLED) performs better than other previous methods.
The submitted paper sounds to be interesting but requires further significant corrections and considerations to become an acceptable work on this journal.
Point 1: As you have many abbreviations, therefore please provide an abbreviation table that on MDPI format must be at the end of the paper, however you made it but is not complete.
Response 1: Thanks very much for your comment. We have provided an abbreviation table at the end of the paper. The table is as follow (Line 490-492):
|
SDLED |
A semi-supervised deep learning approach based on the edge detection network |
|
D-LinkNet |
LinkNet with Pretrained Encoder and Dilated Convolution |
|
VHR |
Very-High-Resolution |
|
CNN |
convolutional neural network |
|
SNNRCE |
self-training nearest neighbor rule using cut edges |
|
WTDS |
weighted ternary decision structure |
|
IoU |
Intersection-over-Union |
|
RCF |
a richer convolutional features network |
|
FER-CNN |
a Faster edge R-CNN |
|
RDP |
Ramer–Douglas–Peucker |
|
HED |
The holistically nested edge detection network |
|
FCN |
a fully convolutional network |
|
DFF |
Dynamic feature fusion |
|
BDCN |
Bi-Directional Cascade Network |
|
SEM |
scale enhancement module |
|
ResNet |
residual network |
|
HSI |
hyperspectral image |
|
CRNN |
convolutional recursive neural network |
|
C-DPMM |
the constrained Dirichlet process mixture model |
|
GPU |
graphics processing unit |
|
TP |
true positives |
|
TN |
true negatives |
|
FP |
false positives |
|
FN |
false negatives |
Point 2: The abstract is too confusing and too short and does not present important points. It should be between 250 to 300 words and concisely mention the problems of previous works and novelties in this paper.
Response 2: Thanks very much for your comment. We have revised the abstract as follow (Line 8-13):
“Automated detection of building in remote sensing images enables grasping the distribution information of buildings in time, which is indispensable for many geographic and social applications such as urban planning, change monitoring and population estimation. The good performance of deep learning in images often depends on a large number of manually labeled samples, which is time-consuming and expensive. Thus, this study focuses on reducing the number of labeled samples used and propose a semi-supervised deep learning approach based on the edge detection network (SDLED), which is the first to introduce semi-supervised learning to the edge detection neural network for extracting building roof boundaries from high-resolution remote sensing images. The approach uses a small number of labeled samples and abundant unlabeled images for joint training. An expert-level semantic edge segmentation model is trained based on labeled samples, which guides unlabeled images to generate pseudo-labels automatically. The inaccurate label sets and manually labeled samples are used to update the semantic edge model together. Particularly, we modified the semantic segmentation network D-LinkNet to obtain high-quality pseudo-labels. Specifically, the main network architecture of D-LinkNet is retained while the multi-scale fusion is added in its second half to improve its performance on edge detection. The SDLED was tested on high spatial resolution remote sensing images taken from Google Earth. Results show that the SDLED performs better than the fully-supervised method. Moreover, when the trained models were used to predict buildings in the neighboring counties, our approach was superior to the supervised way, with the line IoU improvement of at least 6.47% and the F1 score improvement of at least 7.49%.”
Point 3: Similarly, the introduction has very poor structure and lack of literature review. Usually in the chapter of introduction the background and needs of study of for example seismic risk assessment and mapping methods must be highlighted and prepare readers to go further. Then your second chapter should be literature review where you present an overview on the previous works and the main problem statements of work and how it can be improved or overcome on it.
Response 3: Thanks very much for your comment. In the first paragraph of the introduction, we added the problem statement and solution of disaster assessment of buildings to highlight the background and needs of study, the new paragraphs are as follows (line 33-39):
“Automatically extracting building roof from remote sensing images can provide abundant information for urban planning, change detection, disaster assessment and other fields. Taking earthquake monitoring as an example, many approaches have been developed to complete the task of rapidly damage assessments [1,2], which play an important role in rescue and recovery missions. These methods relying on computer technology are effective, which can extract building damage information in a very short time and greatly reduce labor costs.”
In second chapter, we summarize the previous works and draw off the main problem at the end of each paragraph. And the next paragraph is solve the problem of previous paragraph. The end of second paragraph in second chapter maybe not clear, and we revised it as follow (line 147-150):
“However, the traditional deep learning methods often adopted a large number of manually labeled samples to ensure the good performance of deep learning, nevertheless, which is time-consuming and expensive. It is significant to reduce the dependence of network on labeled samples.”
Point 4: Please provide more information about the selected location and data repository and how they have been collected.
Response 4: Thanks very much for your comment. We supplement the data description as follow (line 243-246):
“It is a typical representative of urban architecture, including a variety of typical buildings, which presents different styles in VHR remote sensing images. It is highly representative of the problem of building boundary extraction involved in this study.”
Point 5: Please provide better quality/shape for figures (especially Fig 1 and 2) and provide a systematic format and shape and legend for them. They need more descriptions on the body of paper to explain it in detail not in the figure label.
Response 5: Thanks very much for your comment. Fig 1 and 2 is completed and exported in Visio, and we have replaced them with images with a DPI of 600. And we add more descriptions on the body of paper to explain it now. The description is as follows (line 182-185 and line 220-227):
“Figure 1. Edge-detection-based semi-supervised framework. Labeled dataset is used to train pre-trained model, which predicts unlabeled dataset to get pseudo-labels. Labeled dataset and pseudo-labels are combined to form an extended dataset, which has abundant labeled images. Fine-tuning model is trained with extended dataset based on pre-trained model.”
“Figure 2. Our network for detecting the building edge, which modifies from D-LinkNet. D-LinkNet is composed of encoder, center part and decoder. Part A in our network retains the original encoder and center part from D-LinkNet for effectively extracting the multi-scale features in the image. Part B uses multi-scale fusion to replace the decoder which has a wide receptive field and keeps the ability of spatial detail representation. For each stage of the encoder, we use the transposed convolution layer for up-sampling and restoring the resolution of the feature image to 448 * 448. Then, the network calculates the loss function of each stage to obtain more accurate edge location information.”
Point 6: Results and discussion are not properly organized, and it must show the significant achievements of the proposed method and discuss each table and figure properly and in detail.
Response 6: Thanks very much for your comment. We have reorganized Results and Discussion part. Results is divided into Section 4.4 and Discussion is divided into Section 5. The text in Results and Discussion have revised.
Point :7 It could be great if you do a comparison between your proposed method and some of the available or common other methods to show the efficiency of it.
Response 7: Thanks very much for your comment. We had ompared with traditional methods and some deep learning methods, such as Sobel, Canny, DexiNed and BDCN. Some of these methods just detect the edge of total factor and cannot distinguish building roof, so it can't be measured by our precision evaluation index. Other methods like DexiNed, the performance is not better than BDCN, so we choose the best one for comparison. The results as follow:
|
Sample Scale |
F1_score |
Single IoU |
Whole IoU |
Line IoU (kernel=3) |
Line IoU (kernel=5) |
|
100% (fully- supervised) DexiNed |
0.3034 |
0.2659 |
0.1884 |
0.2070 |
0.2536 |
|
50% (semi- supervised) DexiNed |
0.4549 |
0.4057 |
0.3194 |
0.2976 |
0.3725 |
|
100% (fully- supervised) BDCN |
0. 5362 |
0.4043 |
0. 3941 |
0. 3233 |
0. 3849 |
|
50% (semi- supervised) BDCN |
0. 6267 |
0.4950 |
0. 4866 |
0. 3337 |
0. 4281 |
Point 8: It would be helpful if you provide a general framework or flowchart that how others can implement or use your proposed method for their assessment purposes.
Response 8: Thanks very much for your comment. Our framework is shown in Figure 1. Labeled dataset is used to train pre-trained model, which predicts unlabeled dataset to get pseudo-labels. Labeled dataset and pseudo-labels are combined to form an extended dataset, which has a large number of labeled images. Fine-tuning model is trained with extended dataset based on pre-trained model.
Point 9: In total, your conclusion is good, and you can discuss a bit again about the achievements and novelty of your proposed method.
Response 9: Thanks very much for your comment. We have emphasized the contribution of this paper again in the discussion. The added parts are as follow (line 462-464 and line 471-473):
“This is the first time that semi-supervised learning and edge detection neural network have been combined for extracting building roof boundaries from VHR remote sensing images.”
“In addition, by testing in the surrounding areas, the improvement effect of our method is more obvious, which proves the generalization of our method and provides a new way of thinking for generalization learning.”
Point 10: The paper needs a detailed proofreading and paraphrasing.
Response 10: Thanks very much for your comment. We have rechecked the paper and found some mistakes, and then revised them. For example:
The caption of Figure 5. had some different with original version after typesetting, and we have revised it (line 316-317). “Figure 5. Precision evaluation results for effect of sample scale the supervised method. All the results obtained by our proposed network with the fully-supervised method.”
The same problem appeared in the text of chapter 4.4.1 and we have revised it (line 318-320).” For verifying the influence of samples scale on our proposed network, we trained the network with fully supervised method in an increased amount of training images, respectively the proportions of 25%, 50%, 75% and 100%.”
Point 11: Please make your tables and figures follow a same path and font size and colors and adjust them in a proper way, table 2 is totally disorganized.
Response 11: Thanks very much for your comment. The red and blue fonts are the modification marks according to the modification opinions of the rejected manuscript, which are required in the modification opinions. Now we've removed the previous marks, and remarked it in red according to the comments of you and other reviewers. The table 2 have reorganized (line 329).
Point 12: You did not highlight the problem statement, objectives and novelty of your proposed method; That is why increasing the background of literature review based on the recommended works can help in this manner.
Response 12: Thanks very much for your comment. At the beginning of the introduction, we have added a paragraph to highlight the problem statement, objectives and novelty of proposed method, the new paragraph are as follows (line 33-40):
“Automatically extracting building roof from remote sensing images can provide abundant information for urban planning, change detection, disaster assessment and other fields. Taking earthquake monitoring as an example, many approaches have been developed to complete the task of rapidly damage assessments [1,2], which play an important role in rescue and recovery missions. These methods relying on computer technology are effective, which can extract building damage information in a very short time and greatly reduce labor costs.”
Point 13: Please provide more information about the analysis rather than just F1 score. Show your confusion matrix and the parameters you can achieve from it.
Response 13: Thanks very much for your comment. We had used the IoU(include polygon IoU and line IoU) and F1 score to evaluate the performance of the proposed method. In this paper, only buildings and non-buildings are detected. It is a binary problem, so the confusion matrix may not be suitable for this paper. F1 score and polygon IoU will evaluate the degree of correlation between the real background and the predicted results, while line IoU evaluate the correlation between the real and predicted building boundaries.
Point 14: Overall, the main problem of your paper is the lack of literature review and you can present some new developed machine learning methods and image processing for damage detection, seismic vulnerability, and risk assessment of buildings to attract the attention of readers and show a wide view of your works. Below are some of the recent works, where I found them new and useful to add and make your paper much more interesting:
-ML-EHSAPP: a prototype for machine learning-based earthquake hazard safety assessment of structures by using a smartphone app
-Post-earthquake road damage assessment using region-based algorithms from high-resolution satellite images
-Earthquake Hazard Safety Assessment of Existing Buildings Using Optimized Multi-Layer Perceptron Neural Network
- Multi-Hazard and Spatial Transferability of a CNN for Automated Building Damage Assessment
- Automatic image-based event detection for large-N seismic arrays using a convolutional neural network
PPM-SSNet: An Pyramid Pooling Module-based Semi-Siamese Network for End-to-end Building Damage Assessment from Satellite Imagery
-Seismic loss estimation software: A comprehensive review of risk assessment steps, software development and limitations
-Evaluation of Change Detection Techniques using Very High Resolution Optical Satellite Imagery
Response 14: Thanks very much for your comment. In the first paragraph of the introduction, we added the problem statement and solution of disaster assessment of buildings. Two papers about disaster assessment of buildings are cited.
- Multi-Hazard and Spatial Transferability of a CNN for Automated Building Damage Assessment
-PPM-SSNet: An Pyramid Pooling Module-based Semi-Siamese Network for End-to-end Building Damage Assessment from Satellite Imagery
The new paragraph are as follows (line 33-40):
“Automatically extracting building roof from remote sensing images can provide abundant information for urban planning, change detection, disaster assessment and other fields. Taking earthquake monitoring as an example, many approaches have been developed to complete the task of rapidly damage assessments [1,2], which play an important role in rescue and recovery missions. These methods relying on computer technology are effective, which can extract building damage information in a very short time and greatly reduce labor costs.”

Round 2
Reviewer 3 Report
Dear Authors
The revised manuscript shows a significant improvement. However, as in the previous review mentioned, it would be good to drag the attention of readers to your article and the topic of using machine learning and image processing techniques. To do so, it is necessary to use some related articles and new works to show your wide overview on the topic for instance you can use some of the below works:
-Evaluation of Change Detection Techniques using Very High Resolution Optical Satellite Imagery
-A Review on Application of Soft Computing Techniques for the Rapid Visual Safety Evaluation and Damage Classification of Existing Buildings
-PhyMDAN: Physics-informed knowledge transfer between buildings for seismic damage diagnosis through adversarial learning
Author Response
Dear reviewer,
Thank you very much for your generous help with our paper and the reviewers’ valuable comments concerning our manuscript entitled “Building Extraction from Very-High-Resolution Remote Sensing Images Using Semi-Supervised Semantic Edge Detection” (remotesensing-1216229). We have carefully revised the paper according to these comments. The point-to-point response to the reviewers’ comments is listed below.
p.s. In the revised version, we use the “Track Changes” function to mark up the changes based on your comments. The line number mentioned in the following responses are all collected from our revised manuscript. We restored the revised part of the previous version to black font, which is red font.
Point 1: The revised manuscript shows a significant improvement. However, as in the previous review mentioned, it would be good to drag the attention of readers to your article and the topic of using machine learning and image processing techniques. To do so, it is necessary to use some related articles and new works to show your wide overview on the topic for instance you can use some of the below works:
-Evaluation of Change Detection Techniques using Very High Resolution Optical Satellite Imagery
-A Review on Application of Soft Computing Techniques for the Rapid Visual Safety Evaluation and Damage Classification of Existing Buildings
-PhyMDAN: Physics-informed knowledge transfer between buildings for seismic damage diagnosis through adversarial learning
Response 1: Thanks very much for your comment. We have revised the first paragraph of the introduction and added the above three articles to the reference. The revised paragraph are as follows (Line 33-44):
“In recent years, machine learning and image processing techniques have been serving remote sensing images to mine abundant information for urban planning, change detection, disaster assessment and other fields. Taking earthquake monitoring as an example, soft computing techniques complete seismic vulnerability assessment of existing buildings, which mitigate post-quake effects [1]. Meanwhile, many approaches have been developed to complete the task of rapidly damage assessment [2-4], which play an important role in rescue and recovery missions. The pixel-based change detection is also widely used to accurately analyze the changes of destroyed houses [5]. These methods relying on computer technology are effective, which can extract building damage information in a very short time and greatly reduce labor costs.”

This manuscript is a resubmission of an earlier submission. The following is a list of the peer review reports and author responses from that submission.
Round 1
Reviewer 1 Report
In this paper, the authors address the problem of extracting the edges of buildings from remote images with high-resolution using a semi-supervised learning approach. The employed approach serves to obtain better results than using just manually labelled images. In addition, the model produced in this work deals with the problem of out-of-domain data since it can work with images from regions that were not in the training dataset.
The paper addresses an interesting problem, and the main contribution is the application of a standard semi-supervised learning technique to deal with the lack of annotated data.
From my point of view, there are several issues with this paper that should be tackled before accepting it for publication. My first issue with the paper is that neither the code or the models are freely available. Hence, it is not possible to check the results produced by the authors. In addition, the architecture presented by the authors could be used for other problems; but many interested users might find difficult to implement it. Therefore, it is a must that authors provide their code and their models. Similarly, the dataset employed for this work is not available; hence, it is difficult to reproduce the results.
The second issue is related with Section 5 that should be improved. First, in Section 5.1 the authors only provide a table and a figure, but they do not give any explanation about them. In the rest of the subsections, there are some additional explanations, but it is not clear what the authors are studying, and whether their method is useful. Moreover, in this section, it is not clear how the datasets are employed, since it seems that they are not using the unlabelled data or that they are mixing it with part of the manually labelled dataset. This should be clearly explained in Section 4. Finally,
in this section, the authors keep talking about sample scale, but this concept was not previously explained; so, it is not possible to follow what they are studying.
The final issue is related to the evaluation. The authors are not using a standard dataset, or comparing their method with other existing techniques; so, it is not possible to know whether this approach serve to improve the existing literature for detecting building edges. As an example of a work comparing with previous existing methods, the authors can check the paper available at https://doi.org/10.3390/rs10091496.
Minor comment:
Figure 1. Pseudo-label has a typo.
Author Response
Dear editor,
Thank you very much for your generous help with our paper and the reviewers’ valuable comments concerning our manuscript entitled “Semi-supervised Building Extraction for Very-High-Resolution Remotely Sensed Imagery Based on Semantic Edge Detection” (remotesensing-1076964). We have carefully revised the paper according to these comments. The point to point response to the reviewers’ comments is listed below.
p.s. In the revised version, the red parts are the changes based on the reviewers’ comments. The line number mentioned in the following responses are all collected from our revised manuscript. In addition, Figure 1 have been changed and there may be some typography errors in Line 246-248 and in Line 291-295, which I have revised.
Reviewer 1:
In this paper, the authors address the problem of extracting the edges of buildings from remote images with high-resolution using a semi-supervised learning approach. The employed approach serves to obtain better results than using just manually labelled images. In addition, the model produced in this work deals with the problem of out-of-domain data since it can work with images from regions that were not in the training dataset.
The paper addresses an interesting problem, and the main contribution is the application of a standard semi-supervised learning technique to deal with the lack of annotated data.
From my point of view, there are several issues with this paper that should be tackled before accepting it for publication. My first issue with the paper is that neither the code or the models are freely available. Hence, it is not possible to check the results produced by the authors. In addition, the architecture presented by the authors could be used for other problems; but many interested users might find difficult to implement it. Therefore, it is a must that authors provide their code and their models. Similarly, the dataset employed for this work is not available; hence, it is difficult to reproduce the results.
Response: Thanks very much for your comment. We will open the dataset and part of the code first, and the rest of the code will be open after other related papers are published. The dataset includes three parts of images mentioned in the paper, including 344 labeled images and 1564 unlabeled images of training area, and 288 labeled images of neighboring area. The post-processing code and accuracy evaluation code will open first, and then network code will be open later.
The second issue is related with Section 5 that should be improved. First, in Section 5.1 the authors only provide a table and a figure, but they do not give any explanation about them. In the rest of the subsections, there are some additional explanations, but it is not clear what the authors are studying, and whether their method is useful. Moreover, in this section, it is not clear how the datasets are employed, since it seems that they are not using the unlabelled data or that they are mixing it with part of the manually labelled dataset. This should be clearly explained in Section 4. Finally,
in this section, the authors keep talking about sample scale, but this concept was not previously explained; so, it is not possible to follow what they are studying.
Response: Thanks very much for your comment. In Section 5.1, there may be some mistakes in typesetting. The content of the text has been incorporated into the legend of Figure 5 in Line 291-295. And now it has been modified. The use of datasets is described in Section 3.2 and figure 1. For further explanation, we will continue to explain in Line 235-238.
The final issue is related to the evaluation. The authors are not using a standard dataset, or comparing their method with other existing techniques; so, it is not possible to know whether this approach serve to improve the existing literature for detecting building edges. As an example of a work comparing with previous existing methods, the authors can check the paper available at https://doi.org/10.3390/rs10091496.
Response: Thanks very much for your comment. Our main purpose is to improve the semi supervised learning method, compare the results of full supervised learning and semi supervised learning when using the same neural network. Before, we have done some representative deep learning methods in edge detection, such as RCF and DexiNed. But there are some problems with their results (building boundary is not closed), which are not conducive to precision calculation, so they are not included in the paper. The following is the experimental results of RCF and DexiNed. (The comparison chart is in the Word "Response to Reviewer1.docx".)
Minor comment:
Figure 1. Pseudo-label has a typo.
Response: Thanks very much for pointing out the text error. ‘Pesudl’ has been changed to ‘Pseudo’, in Line 172, Figure 1.

Reviewer 2 Report
I have reviewed the manuscript entitled 'Semi-supervised Building Extraction for Very-High-Resolution Remotely Sensed Imagery Based on Semantic Edge Detection'. The manuscript presents an extensive deep learning approach using mainly Google Earth imagery.
My major concern is related to the fact that the authors did not present any Discussion of their results. Their 'Results and Discussion' chapter is merely descriptive. In scientific article the results should be compared to other scientific works. Therefore, the manuscript could be considered for publication again in case they can extend it an appropriate Discussion chapter.
My additional comments are the following:
- I suggest to merge Chapter 1 and 2 since they are both discussing the background.
- Figure 1. What is a 'Pesdul-layer'?
- How did the authors do the image acquisition from Google Earth? Why they were not using georeferenced imagery?
- The following Figures are not referred to in the body text: Figure 7, 9.
- The caption of Figure 5 seems to be a bit long. Please review it and if some part of it belongs to the body text then modify it. Also in the same caption the authors wrote 'Table 25' which seems to be also corrected.
- I also suggest the authors to read and possibly cite the following literature to the introduction chapter if they found them useful:
-
- Hossain, M.D.; Chen, D. (2019). Segmentation for Object-Based Image Analysis (OBIA): A review of algorithms and challenges from remote sensing perspective, ISPRS Journal of Photogrammetry and Remote Sensing 150, 115-134.
- Schlosser, A.D.; Szabó, G.; Bertalan, L.; Varga, Z.; Enyedi, P.; Szabó, S. (2020). Building Extraction Using Orthophotos and Dense Point Cloud Derived from Visual Band Aerial Imagery Based on Machine Learning and Segmentation. Remote Sensing 12, 2397.
- Zhang, X.; Cui, J.; Wang, W.; Lin, C. A Study for Texture Feature Extraction of High-Resolution Satellite Images Based on a Direction Measure and Gray Level Co-Occurrence Matrix Fusion Algorithm (2017). Sensors 17, 1474.
Author Response
Dear editor,
Thank you very much for your generous help with our paper and the reviewers’ valuable comments concerning our manuscript entitled “Semi-supervised Building Extraction for Very-High-Resolution Remotely Sensed Imagery Based on Semantic Edge Detection” (remotesensing-1076964). We have carefully revised the paper according to these comments. The point to point response to the reviewers’ comments is listed below.
p.s. In the revised version, the red parts are the changes based on the reviewers’ comments. The line number mentioned in the following responses are all collected from our revised manuscript. In addition, Figure 1 have been changed and there may be some typography errors in Line 246-248 and in Line 291-295, which I have revised.
Reviewer 2:
I have reviewed the manuscript entitled 'Semi-supervised Building Extraction for Very-High-Resolution Remotely Sensed Imagery Based on Semantic Edge Detection'. The manuscript presents an extensive deep learning approach using mainly Google Earth imagery.
My major concern is related to the fact that the authors did not present any Discussion of their results. Their 'Results and Discussion' chapter is merely descriptive. In scientific article the results should be compared to other scientific works. Therefore, the manuscript could be considered for publication again in case they can extend it an appropriate Discussion chapter.
Response: Thanks very much for your comment. Our main purpose is to improve the semi supervised learning method, compare the results of full supervised learning and semi supervised learning when using the same neural network. Before, we have done some representative deep learning methods in edge detection, such as RCF and DexiNed. But there are some problems with their results (building boundary is not closed), which are not conducive to precision calculation, so they are not included in the paper. The following is the experimental results of RCF and DexiNed.(The comparison chart is in the attachment Word "Response to Reviewer1.docx".)
My additional comments are the following:
- I suggest to merge Chapter 1 and 2 since they are both discussing the background.
Response: Thanks very much for your comment. The first chapter is an overall description of the background, while the second chapter is a detailed description of the existing methods, with different emphasis.
- Figure 1. What is a 'Pesdul-layer'?
Response: Thanks very much for your comment. ‘Pesdul-label’ is misspelled and has been changed to ‘pseudo-label’ in Line 172, Figure 1. The unlabeled images are predicted by the pre-trained model to get the pseudo-label.
- How did the authors do the image acquisition from Google Earth? Why they were not using georeferenced imagery?
Response: Thanks very much for your comment. We obtained the image from the Imagesky in Suzhou which is a Chinese company. We explained the origin of the picture in acknowledgments in Line 416-417.
- The following Figures are not referred to in the body text: Figure 7, 9.
Response: Thanks very much for your comment. Figures 7 and 9 show the comparison of experimental results, which have been cited in Line 322-323 and Line 365.
- The caption of Figure 5 seems to be a bit long. Please review it and if some part of it belongs to the body text then modify it. Also in the same caption the authors wrote 'Table 25' which seems to be also corrected.
Response: Thanks very much for your comment. In Section 5.1, there may be some mistakes in typesetting. The content of the text has been incorporated into the legend of Figure 5 in Line 291-306. And now it has been modified. The use of datasets is described in Section 3.2 and figure 1. For further explanation, we will continue to explain in Line 235-238 “The first part of the samples is used for training pre-model. The second part of the samples is added into the first part after prediction by pre-model. The two parts samples are forms extended dataset for training fine-tuning model. The third part of the samples is only used for the generalization ability test.”.
- I also suggest the authors to read and possibly cite the following literature to the introduction chapter if they found them useful:
- Hossain, M.D.; Chen, D. (2019). Segmentation for Object-Based Image Analysis (OBIA): A review of algorithms and challenges from remote sensing perspective, ISPRS Journal of Photogrammetry and Remote Sensing 150, 115-134.
- Schlosser, A.D.; Szabó, G.; Bertalan, L.; Varga, Z.; Enyedi, P.; Szabó, S. (2020). Building Extraction Using Orthophotos and Dense Point Cloud Derived from Visual Band Aerial Imagery Based on Machine Learning and Segmentation. Remote Sensing 12, 2397.
- Zhang, X.; Cui, J.; Wang, W.; Lin, C. A Study for Texture Feature Extraction of High-Resolution Satellite Images Based on a Direction Measure and Gray Level Co-Occurrence Matrix Fusion Algorithm (2017). Sensors 17, 1474.
Response: Thanks very much for your comment. After reading the three articles, they are cited in the chapter 1 in Line 35 and Line 45, and in the chapter 2 in Line 120.

Reviewer 3 Report
Semi-supervised building extraction for very-high-resolution remotely sensed imagery based on semantic edge detection.
The authors are proposing a semi-supervised semantic edge detection method based on deep learning to extract building edges from very-high-resolution remote sensing images. The purpose of the work was to address the intrinsic limitations of incomplete boundaries, insufficient labelled dataset, and low precision of some existing methods for building extraction. The authors have considered a practical problem that is essential for building description. However, I would like the following concerns to be adequately addressed for the manuscript to be recommended for publication.
- The topic is confusing in meaning and does not adequately capture the content of the manuscript. I suggest that it be changed to something like “Building extraction from very-high-resolution remote sensing images using semi-supervised semantic edge detection deep learning”.
- Considering the review of literature as in Section 2, what is the missing void that the authors are trying to fill? The authors should provide this information at the end of the review to strengthen the contributions of the work.
- The statement “Our loss function [10] can be formulated as…” should be replaced by “Our loss function based on [10] can be formulated as…” because reference [10] is not the work of the authors.
- There are some language typos to be corrected in the manuscript, for example “FP (false positions)” on page 9 should be “FP (false positives)”.
- The authors should be consistent with the use of decimal places because 3 and 4 decimal places are used in Tables 1 to 4. I suggest the use of 4 decimal places throughout by appending 0 to the end of 3 decimal places.
- It is not clear why accuracy was used in the manuscript, which was not introduced. The performance metrics discussed are IoU, precision, recall and F1 score. Accuracy metric should be included or change to precision throughout the manuscript.
- The authors only compared the results of their method with fully supervised building edge detection++, why not compared their results with results computed by some of the existing reviewed methods, for instance references [14], [22] and [30]?
Author Response
Dear editor,
Thank you very much for your generous help with our paper and the reviewers’ valuable comments concerning our manuscript entitled “Semi-supervised Building Extraction for Very-High-Resolution Remotely Sensed Imagery Based on Semantic Edge Detection” (remotesensing-1076964). We have carefully revised the paper according to these comments. The point to point response to the reviewers’ comments is listed below.
p.s. In the revised version, the red parts are the changes based on the reviewers’ comments. The line number mentioned in the following responses are all collected from our revised manuscript. In addition, Figure 1 have been changed and there may be some typography errors in Line 246-248 and in Line 291-295, which I have revised.
Reviewer 3
Semi-supervised building extraction for very-high-resolution remotely sensed imagery based on semantic edge detection.
The authors are proposing a semi-supervised semantic edge detection method based on deep learning to extract building edges from very-high-resolution remote sensing images. The purpose of the work was to address the intrinsic limitations of incomplete boundaries, insufficient labelled dataset, and low precision of some existing methods for building extraction. The authors have considered a practical problem that is essential for building description. However, I would like the following concerns to be adequately addressed for the manuscript to be recommended for publication.
- The topic is confusing in meaning and does not adequately capture the content of the manuscript. I suggest that it be changed to something like “Building extraction from very-high-resolution remote sensing images using semi-supervised semantic edge detection deep learning”.
Response: Thanks very much for your comment. We changed the theme to "building extraction from very high resolution remote sensing images using semi supervised semantic edge detection deep learning" in Line 2-3.
- Considering the review of literature as in Section 2, what is the missing void that the authors are trying to fill? The authors should provide this information at the end of the review to strengthen the contributions of the work.
Response: Thanks very much for your comment. At the end of Section 2.1, 2.2 and 2.3, we add some information to strengthen the contributions of the work in Line 119-120 (“The purpose of segmentation is not only satisfied with pixel level assistance, lies in the object identification with accurate boundary [10].”) and Line 139-141 (“However, deep learning needs a large number of manually labeled samples to train model better. It will be helpful to reduce the sample requirement in deep learning.”).
- The statement “Our loss function [10] can be formulated as…” should be replaced by “Our loss function based on [10] can be formulated as…” because reference [10] is not the work of the authors.
Response: Thanks very much for your comment. The original sentence is changed to “Our loss function based on [10] can be formulated as ” in Line 201
- There are some language typos to be corrected in the manuscript, for example “FP (false positions)” on page 9 should be “FP (false positives)”.
Response: Thanks very much for your comment. This is my language typo. “FP (false positions)” has been changed to “FP (false positions)” in Line 276
- The authors should be consistent with the use of decimal places because 3 and 4 decimal places are used in Tables 1 to 4. I suggest the use of 4 decimal places throughout by appending 0 to the end of 3 decimal places.
Response: Thanks very much for your comment. It's my fault that I didn't keep the decimal places consistent. Now, all the decimal places are changed to 4 in Tables 1 to 4.
- It is not clear why accuracy was used in the manuscript, which was not introduced. The performance metrics discussed are IoU, precision, recall and F1 score. Accuracy metric should be included or change to precision throughout the manuscript.
Response: Thanks very much for your comment. We have used too much subjective evaluation in this part, which is inappropriate and has been revised in Line 357- Line 358 (“Compared with the training area, the values of all the evaluating indicators decreased obviously.”).
- The authors only compared the results of their method with fully supervised building edge detection++, why not compared their results with results computed by some of the existing reviewed methods, for instance references [14], [22] and [30]?
Response: Thanks very much for your comment. Thanks very much for your comment. Our main purpose is to improve the semi supervised learning method, compare the results of full supervised learning and semi supervised learning when using the same neural network. Before, we have done some representative deep learning methods in edge detection, such as RCF and DexiNed. But there are some problems with their results (building boundary is not closed), which are not conducive to precision calculation, so they are not included in the paper. The following is the experimental results of RCF and DexiNed. (The comparison chart is in the attachment Word "Response to Reviewer1.docx".)

Round 2
Reviewer 1 Report
The authors claim that they will release the dataset and code of their work, but they did not do it. There is no link or information about where the code and dataset are available. This should be done before considering the paper for publication.
Section 5 should still be improved since it is difficult to follow what the authors are studying at each subsection.
Finally, a discussion related to other papers is still needed. It is true that the authors are focused on improving the semi supervised learning method, but do this have an impact in the context of building extraction? If this method is far from the results obtained by other methods, I still wonder what is the real contribution since pseudolabeling is a standard semi-supervised learning technique.
Author Response
Dear editor,
Thank you very much for your generous help with our paper and the reviewers’ valuable comments concerning our manuscript entitled “Semi-supervised Building Extraction for Very-High-Resolution Remotely Sensed Imagery Based on Semantic Edge Detection” (remotesensing-1076964). We have carefully revised the paper according to these comments. The point to point response to the reviewers’ comments is listed below.
p.s. In the revised version, the red parts are the changes based on the reviewers’ comments. The line number mentioned in the following responses are all collected from our revised manuscript.
Reviewer 1(Round 2):
The authors claim that they will release the dataset and code of their work, but they did not do it. There is no link or information about where the code and dataset are available. This should be done before considering the paper for publication.
Response: Thanks very much for your comment. We have open the post-processing code and accuracy evaluation code to github in https://github.com/lovexixuegui/SDLED. And we send the dataset to Baidu cloud in https://pan.baidu.com/s/1bSLz4pdNVgmRmdBDcre6xQ with the extraction code f1tx. The rest of the code will be open after other related papers are published.
Section 5 should still be improved since it is difficult to follow what the authors are studying at each subsection.
Response: Thanks very much for your comment. Section 5.1 is about the influence of the number of samples on training the edge detection model. The conclusion is that the more the quantity, the better the effect. According to this conclusion, we discuss in section 5.2 whether increasing the number of unlabeled samples can also improve the accuracy. Because the unlabeled samples will generate pseudo-labels with our method, which improve the number of samples. In section 5.3, we test our method in the neighboring districts for generalization ability. In section 5.4, we test a sample selection method to screen pseudo-labels which is more complete, but the method seems useless. And we re-edit section 5 for easier to understand in Line 298-299 and Line 339-345.
Finally, a discussion related to other papers is still needed. It is true that the authors are focused on improving the semi supervised learning method, but do this have an impact in the context of building extraction? If this method is far from the results obtained by other methods, I still wonder what is the real contribution since pseudolabeling is a standard semi-supervised learning technique.
Response: Thanks very much for your comment. We compare our method with DexiNed, which is also a edge detection network. The results and discussion in paper Line 306-345.

Reviewer 2 Report
Dear Authors,
The manuscript has been improved in almost all cases however I still miss a more clear discussion section where your results are compared with other already published references. I still believe that this part should be important to include.
Author Response
Dear editor,
Thank you very much for your generous help with our paper and the reviewers’ valuable comments concerning our manuscript entitled “Semi-supervised Building Extraction for Very-High-Resolution Remotely Sensed Imagery Based on Semantic Edge Detection” (remotesensing-1076964). We have carefully revised the paper according to these comments. The point to point response to the reviewers’ comments is listed below.
p.s. In the revised version, the red parts are the changes based on the reviewers’ comments. The line number mentioned in the following responses are all collected from our revised manuscript.
Reviewer 2(Round 2):
The manuscript has been improved in almost all cases however I still miss a more clear discussion section where your results are compared with other already published references. I still believe that this part should be important to include.
Response: Thanks very much for your comment. Section 5.1 is about the influence of the number of samples on training the edge detection model. The conclusion is that the more the quantity, the better the effect. According to this conclusion, we discuss in section 5.2 whether increasing the number of unlabeled samples can also improve the accuracy. Because the unlabeled samples will generate pseudo-labels with our method, which improve the number of samples. In section 5.3, we test our method in the neighboring districts for generalization ability. In section 5.4, we test a sample selection method to screen pseudo-labels which is more complete, but the method seems useless. And we re-edit section 5 for easier to understand in Line 298-299 and Line 339-345.. And we compare our method with DexiNed, which is also a edge detection network. The results and discussion in paper Line 306-345.
